# Rhabdo-immunodeficiency virus, a murine model of acute HIV-1 infection

Rachel A Liberatore[1,2†], Emily J Mastrocola[1,2], Elena Cassella[1], Fabian Schmidt[1], Jessie R Willen[1,2], Dennis Voronin[1‡], Trinity M Zang[1,2], Theodora Hatziioannou[1], Paul D Bieniasz[1,2]*

[1]Laboratory of Retrovirology, The Rockefeller University, New York, United States; [2]Howard Hughes Medical Institute, The Rockefeller University, New York, United States

**Abstract** Numerous challenges have impeded HIV-1 vaccine development. Among these is the lack of a convenient small animal model in which to study antibody elicitation and efficacy. We describe a chimeric Rhabdo-Immunodeficiency virus (RhIV) murine model that recapitulates key features of HIV-1 entry, tropism and antibody sensitivity. RhIVs are based on vesicular stomatitis viruses (VSV), but viral entry is mediated by HIV-1 Env proteins from diverse HIV-1 strains. RhIV infection of transgenic mice expressing human CD4 and CCR5, exclusively on mouse CD4+ cells, at levels mimicking those on human CD4+ T-cells, resulted in acute, resolving viremia and CD4+ T-cell depletion. RhIV infection elicited protective immunity, and antibodies to HIV-1 Env that were primarily non-neutralizing and had modest protective efficacy following passive transfer. The RhIV model enables the convenient in vivo study of HIV-1 Env-receptor interactions, antiviral activity of antibodies and humoral responses against HIV-1 Env, in a genetically manipulatable host.

**\*For correspondence:**
pbieniasz@rockefeller.edu

**Present address:** †RenBio, New York, United States; ‡Regeneron, New York, United States

**Competing interests:** The authors declare that no competing interests exist.

## Introduction

Numerous challenges have impeded the development of a vaccine that protects against HIV-1 infection. Perhaps the most important of these are intrinsic obstacles to the elicitation of antibodies that recognize the trimeric HIV-1 envelope (Env) spike and inhibit viral replication (*Burton and Mascola, 2015*; *Escolano et al., 2017*). Large portions of the HIV-1 Env trimer are conformationally flexible and shielded by glycosylation, and such properties inhibit recognition by antibodies (*Burton and Mascola, 2015*; *Escolano et al., 2017*). Additionally, large HIV-1 in vivo population sizes and short generation times, accompanied by error prone replication (~$10^{-4}$/base/cycle) and recombination, means that vast numbers of sequence variants are generated every day in each infected individual (*Coffin, 1995*). Thus, while infected individuals generate strain-specific neutralizing antibodies that impose selective pressure on viral populations and influence viral sequence evolution (*Richman et al., 2003*; *Wei et al., 2003*), the discrepant evolutionary rates that characterize HIV-1 Env and antibody-generating B-cells ensures that antibodies present in each individual are generally poorly effective against contemporaneous autologous viruses (*Richman et al., 2003*; *Wei et al., 2003*). Moreover, HIV-1 sequence diversification has occurred in millions of individual humans over approximately 100 years, yielding vast global diversity of HIV-1 Env proteins (*Korber and Gnanakaran, 2009*). This large and evolving population of HIV-1 Env proteins with intrinsic antibody evasion mechanisms makes the elicitation of broadly effective antibodies by vaccines a formidable task (*Mascola and Haynes, 2013*). Nevertheless, rare HIV-1 infected individuals generate potent, broadly neutralizing antibodies (bNAbs) that are capable of neutralizing many circulating HIV-1 strains (*Klein et al., 2013*; *Wu et al., 2010*). However, they typically arise only after years of infection (*Landais and Moore, 2018*). The breadth with which bNAbs neutralize HIV-1 strains is likely a function of their rarity, as any frequently occurring bNAbs would drive frequent

**eLife digest** One of the main obstacles to developing a vaccine against HIV-1 is teaching the immune system to recognize the envelope proteins on the surface of the virus, which are also found on infected cells. Envelope proteins allow HIV-1 to attach to and infect a type of human immune cell known as a T-cell, by interacting with proteins on its membrane called CD4 and CCR5.

Antibodies are proteins produced by the immune system that can stop HIV-1 from spreading. They can recognize and attach to envelope proteins, thus tagging infected cells so the immune system can attack them, and 'neutralizing' viral particles to prevent them from infecting more cells. To make a vaccine against HIV-1, scientists need to teach the immune system how to make neutralizing antibodies. Unfortunately, HIV-1 only replicates in humans and chimpanzees, making it difficult to study how these antibodies are generated.

Now, Liberatore et al. have developed a hybrid virus that recreates key features of HIV-1 infection in mice. The interior of these viruses is made up of components from a rhabdovirus, which replicates well in mice, with envelope proteins from HIV-1 incorporated into the viruses' exterior. Therefore, despite having different replication machinery, these hybrid viruses – nicknamed 'RhIV' – are able to infect the cells of mice using the same attachment mechanism as HIV-1.

Next, Liberatore et al. genetically modified mice to produce human CD4 and CCR5 proteins, so RhIV could attach to their T-cells and get inside. The virus rapidly killed the cells it infected, similar to early HIV-1 infection in humans. But, unlike HIV-1 infection in humans, the mice were able to get rid of the virus within a couple of weeks. When the mice were exposed to RhIV a second time, they were partially protected against re-infection. This 'vaccine effect' was even stronger if the mice were exposed a third time, making them almost immune to the virus. However, the effect could not be attributed exclusively to antibodies, since mice unable to make antibodies still gained some immune protection after infection with RhIV.

The results showed that antibodies produced by the infected mice could recognize HIV-1 envelope proteins, but were unable to neutralize viral particles. Nevertheless, transferring antibodies from infected mice after recovery into healthy mice that had never been exposed to the virus partially protected the healthy mice from infection.

This new model system for HIV-1 infection should make it easier to test new types of vaccines in a context where the vaccinated animal can be challenged with RhIV. Additionally, the ability to genetically engineer both the virus and the mouse host – for example by making mice that produce human antibodies – allows further studies into the development of antibodies that recognize the HIV-1 envelope.

resistance (i.e. loss of activity against circulating strains). What distinguishes the rare individuals who generate bNAbs, and whether it is possible to generate bNAbs in a significant fraction of humans through vaccination, are key issues confronting the HIV-1 vaccine research field (*Landais and Moore, 2018*).

Another significant impediment to HIV-1 vaccine development is the availability of a convenient animal model system in which to study antibody elicitation and efficacy (*Hatziioannou and Evans, 2012*). HIV-1 host range is confined to humans and chimpanzees, severely curtailing options for testing vaccines and other prevention strategies. To partly circumvent this problem, chimeric retroviruses based on simian immunodeficiency virus (SIV) that express HIV-1 Env proteins, termed (simian HIVs or SHIVs), have been developed (*Li et al., 1992*; *Luciw et al., 1995*). Following engineering and a period of adaptation to overcome the sub-optimal use of macaque CD4 receptors, these viruses can often replicate persistently and cause AIDS-like disease in macaques (*Del Prete et al., 2017*; *Joag et al., 1996*). Additionally, particular minimally modified HIV-1 strains have been adapted to replicate in pig-tailed macaques (*Hatziioannou et al., 2014*; *Schmidt et al., 2019*). Though useful, these models require the significant resources associated with investigations in primates. Small animal models based on immunodeficient mice engrafted with human cells and tissues have provided an experimental system for the in vivo testing of antibodies and other molecules as preventative agents (*Hatziioannou and Evans, 2012*; *Mosier et al., 1991*; *Namikawa et al., 1988*; *Watanabe et al., 2007*; *Wege et al., 2008*). However, these models are also inconvenient and

costly, and human cell engrafted mice generate weak and inconsistent immune responses. Moreover, an important advantage of mouse models, that is the ability to genetically manipulate their immune systems, is lost when the viral target cells and immune system are derived from a human graft.

Here, we describe the development of a virus/host animal model that incorporates the critical feature of the HIV-1 viral particle (the Env spike), that is the target of antiviral antibodies, and recapitulates key features of HIV-1 entry and tissue tropism. Specifically, we generated recombinant derivatives of the rhabdovirus, vesicular stomatitis virus (VSV), in which the native envelope glycoprotein (G) is replaced by HIV-1 Env from various subtypes, including transmitted founder strains. In these Rhabdo-Immunodeficiency viruses (RhIV), replication is entirely dependent on HIV-1 Env, as well as human CD4 and coreceptors on target cells. In parallel, we constructed transgenic mice that express human CD4 and CCR5, exclusively in mouse CD4-positive cells, at levels mimicking those on human CD4+ T-cells. Infection of these transgenic mice with RhIVs results in rapid, specific depletion of CD4+ T-cells and an acute viremia that resolves, followed by development of antibodies directed against the HIV-1 envelope. The RhIV model thus enables the convenient in vivo study of HIV-1 Env-receptor interactions, and their inhibition by antibodies in a genetically manipulatable host.

## Results

### Generation and characterization of Rhabdo-Imunodeficiency viruses (RhIV)

HIV-1 itself cannot replicate in murine cells, even when they are engineered to expressed human versions of HIV-1 receptors and essential Tat cofactors that enable HIV-1 entry and transcription in murine cells (*Bieniasz and Cullen, 2000*; *Mariani et al., 2000*). However, VSV has a very broad tropism and numerous VSV-based chimeric viruses expressing functional, heterologous envelope proteins, including that of HIV-1, have been generated (*Boritz et al., 1999*; *Johnson et al., 1997*; *Rabinovich et al., 2014*; *Rose et al., 2001*). We generated a panel of chimeric Rhabdo-Immunodeficiency viruses (RhIVs) encoding Env proteins from diverse HIV-1 strains including those representative of strains circulating in human populations. Initially we constructed RhIV strains encoding the subtype B Env proteins of HIV-1$_{NL4-3}$, a laboratory adapted X4-tropic strain, HIV-1$_{ADA}$, a macrophage tropic primary isolate and HIV-1$_{AD17}$, an R5 tropic transmitted founder (T/F) HIV-1 strain. The ectodomain and transmembrane domains of HIV-1 Env were fused to the cytoplasmic tail of VSV-G (*Figure 1A*). Retaining the VSV-G cytoplasmic tail has the potential to perturb the tertiary structure of the HIV-1 ectodomain, but this strategy improves HIV-1 Env incorporation into VSV particles (*Johnson et al., 1997*). RhIV$_{NL4-3}$, RhIV$_{ADA}$, and RhIV$_{AD17}$ strains were rescued from recombinant DNA, plaque purified and expanded. Western blot analysis of RhIV$_{ADA}$ showed that the HIV-1 envelope protein was readily detectable in lysates from infected cells and pelleted virions (*Figure 1B*). RhIV$_{NL4-3}$ and RhIV$_{AD17}$ strains displayed the appropriate receptor specificity when used to infect GHOSTX4 or GHOSTR5 cells (*Figure 1C*) and infection of these cells with RhIV, but not VSV, led to the appearance of large syncytia, in addition to a pronounced cytopathic effect (*Figure 1C*). We therefore generated a larger panel of RhIV constructs expressing a variety of physiologically relevant envelope proteins from HIV-1 clades A, B, and C, including T/F viruses. All of these viruses replicated well in vitro, reaching titers of ~$10^6$ to $10^7$ plaque forming units (PFU)/ml (*Figure 1D*).

To facilitate the monitoring of RhIV infection, we generated a RhIV$_{AD17}$ based reporter virus that encoded GFP (*Figure 1E*). RhIV$_{AD17}$(GFP) replicated robustly in GHOSTR5 cells, generating green fluorescent syncytia, as well as in a human T-cell line MT2/R5 engineered to express the CCR5 coreceptor (*Figure 1F and G*). Live imaging of RhIV$_{AD17}$(GFP) replication in 293T/CD4/CCR5 cell monolayers suggested a dominant mode of viral spread in cell monolayers via direct cell-cell transmission, with additional viral transmission to distal cells (*Video 1*). We also generated RhIV strains expressing nanoluciferase (nLuc, *Figure 1E*). Infection of TZMbl cells (a popular target cell for HIV-1 neutralization assays) with RhIV$_{AD17}$(nLuc) generated high levels of nLuc within a few hours of infection (*Figure 1H*). Analysis of a panel of RhIV (nLuc) and corresponding HIV-1 (nLuc) viruses revealed that sensitivity to the CCR5-binding antagonist maraviroc was similar for each HIV-1 envelope in the context of either HIV-1 or RhIV infection (*Figure 1I*).

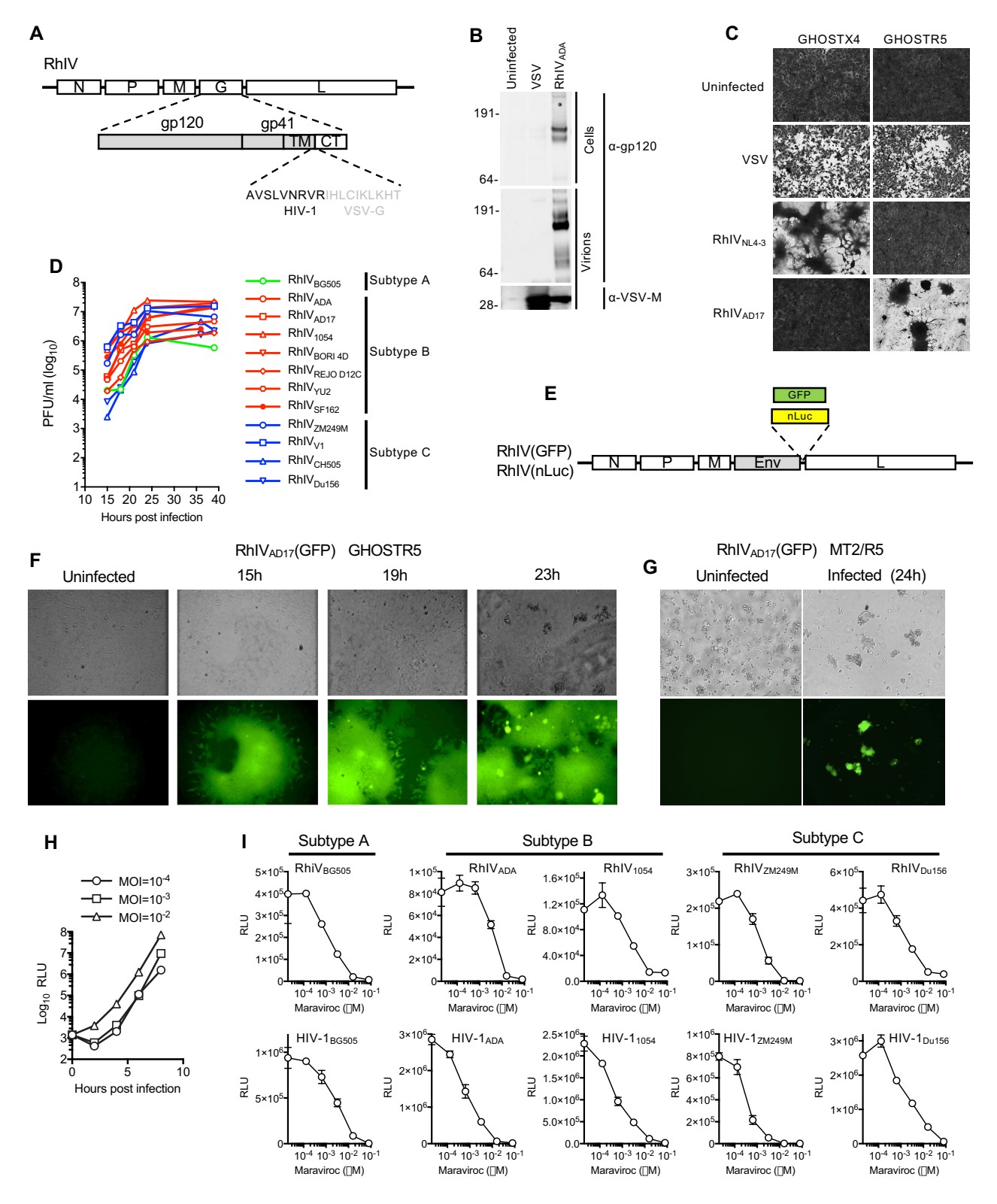

**Figure 1.** Characterization and in vitro replication properties of RhIV strains. (**A**) Schematic representation of RhIV genomes in which VSV-G ectodomain and transmembrane sequences are replaced with HIV-1 Env counterparts. (**B**) Western blot analysis of HIV-1 gp160/120 and VSV-M protein levels in RhIV infected cells and extracellular virions. (**C**) Monolayers of GHOST-X4 or GHOST-R5 cells stained with crystal violet 24 hr after infection with VSV or RhIV strains. (**D**) Yield of various RhIV strains in plaque forming units/ml (PFU/ml) during replication in 293 T/CD4/CCR5 cells. (**E**) Schematic

*Figure 1 continued on next page*

*Figure 1 continued*

representation of RhIV genomes in which a GFP or nanoluciferase (nLuc) reporter is included. (F,G) Micrographs of GHOST-R5 (F) and MT4-R5 (G) at the indicated times after infection with RhIV$_{AD17}$(GFP). (H) Luciferase expression in TZM-Bl cells over time following infection with RhIV$_{AD17}$(nLuc) at the indicated MOIs. (I) Inhibition of RhIV(nLuc) and corresponding HIV-1(nLuc) strains by the CCR5 inhibitor, Maraviroc.

## Neutralization properties of RhIV virions

We compared the sensitivity of RhIV (nLuc) and HIV-1 (nLuc) viruses carrying various HIV-1 Env proteins to neutralization by a panel of well characterized bNAbs. The panel targeted various epitopes on the HIV-1 envelope: PG16 and PG9 recognize a quaternary epitope at the apex of the envelope trimer formed by the V2 loop (*Walker et al., 2009*), 10–1074 recognizes a glycosylation dependent epitope in the V3 loop (*Mouquet et al., 2012*), VRC01 and 3BNC117 target the CD4-binding site (*Scheid et al., 2011*; *Wu et al., 2010*) and 10E8 targets an epitope in the membrane proximal external region (MPER) (*Huang et al., 2012*). In general, matched RhIV (nLuc) and HIV-1 (nLuc) viruses exhibited similar neutralization properties (*Figure 2A and B*). While most RhIV and HIV-1 strains were sensitive to the quaternary epitope targeting PG16 and PG9 antibodies, the RhIV$_{1054}$/HIV-1$_{1054}$, RhIV$_{SF162}$/HIV-1$_{SF162}$ and RhIV$_{V1}$/HIV-1$_{V1}$ virus pairs each shared the property of being resistant to PG9 and PG16 (*Figure 2A and B*). The HIV-1$_{V1}$ strain was unusual in exhibiting near complete resistance to all bNAbs tested, except 10–1074 and this property was preserved in the corresponding RhIV$_{V1}$ chimeric virus (*Figure 2B*). There were, nevertheless, some discrepancies in the potencies with which matched HIV-1 (nLuc) and RhIV (nLuc) viruses were neutralized by bNAbs. In one example, the MPER-targeting bNAb (10E8) neutralized RhIV$_{CH505}$ but did not neutralize HIV-1$_{CH505}$ (*Figure 2B*). These occasional discrepancies may be the result of the HIV-1 and VSV-G cytoplasmic tails imposing different conformations on the Env ectodomain. Alternatively there may be differences in Env spike density, heterogeneity and distribution on RhIV virions as compared to HIV-1 virions.

## Transgenic mice expressing HIV-1 receptors

To generate small animals that had the potential of being infected by RhIVs, we generated transgenic mice expressing human CD4 (hCD4) along with the CCR5 coreceptor. We engineered a construct that contained the murine *Cd4* promoter and intron driving expression of human *CD4* and *CCR5* cDNAs separated by sequences encoding an FMDV 2A site (*Figure 3A*) (*Seay et al., 2013*), with the goal of ensuring that hCD4 would be present exclusively on murine CD4+ cells, and tight linkage between human *CD4* and *CCR5* expression.

Analysis of several independent transgenic mouse lines revealed variable levels of cell surface hCD4. We selected three transgenic mouse lines, A1, C18 and B4 that had high, intermediate and low levels of hCD4 expression respectively (*Figure 3B*). The A1 line mimicked the levels of hCD4 found on human CD4+ T-cells (*Figure 3C*) and was used in subsequent experiments unless otherwise indicated. Levels of CCR5 (as indicated by fluorescence intensity) on the CD4+ cells in the A1 mice were also similar to levels of CCR5 on human CD4+ cells. However, as expected ~100% of hCD4+ cells in the blood of A1 mice were CCR5+ (*Figure 3D*), while the fraction of CD4+ T-cells that also express CCR5 is known to vary according to

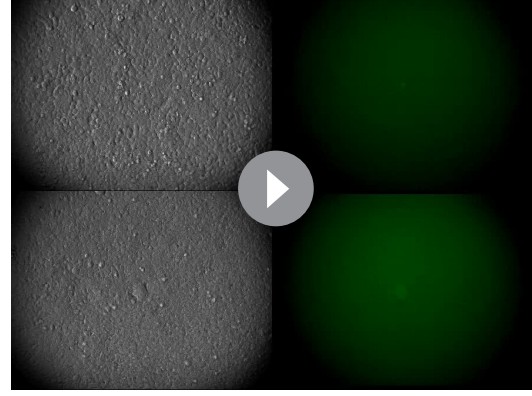

**Video 1.** Spreading replication of RhIV$_{AD17}$(GFP). Cells (293 T/CD4/CCR5) were infected with RhIVAD17(GFP) at low MOI (0.0001) and placed in VivaView FL incubator fluorescence microscope imaging system (Olympus). At 6 hr after infection, individual GFP positive cells were identified and centered in a field of observation and images acquired every 5 min thereafter. The movie represents 24 hr of observation (from 6 hr to 30 hr after infection).
https://elifesciences.org/articles/49875#video1

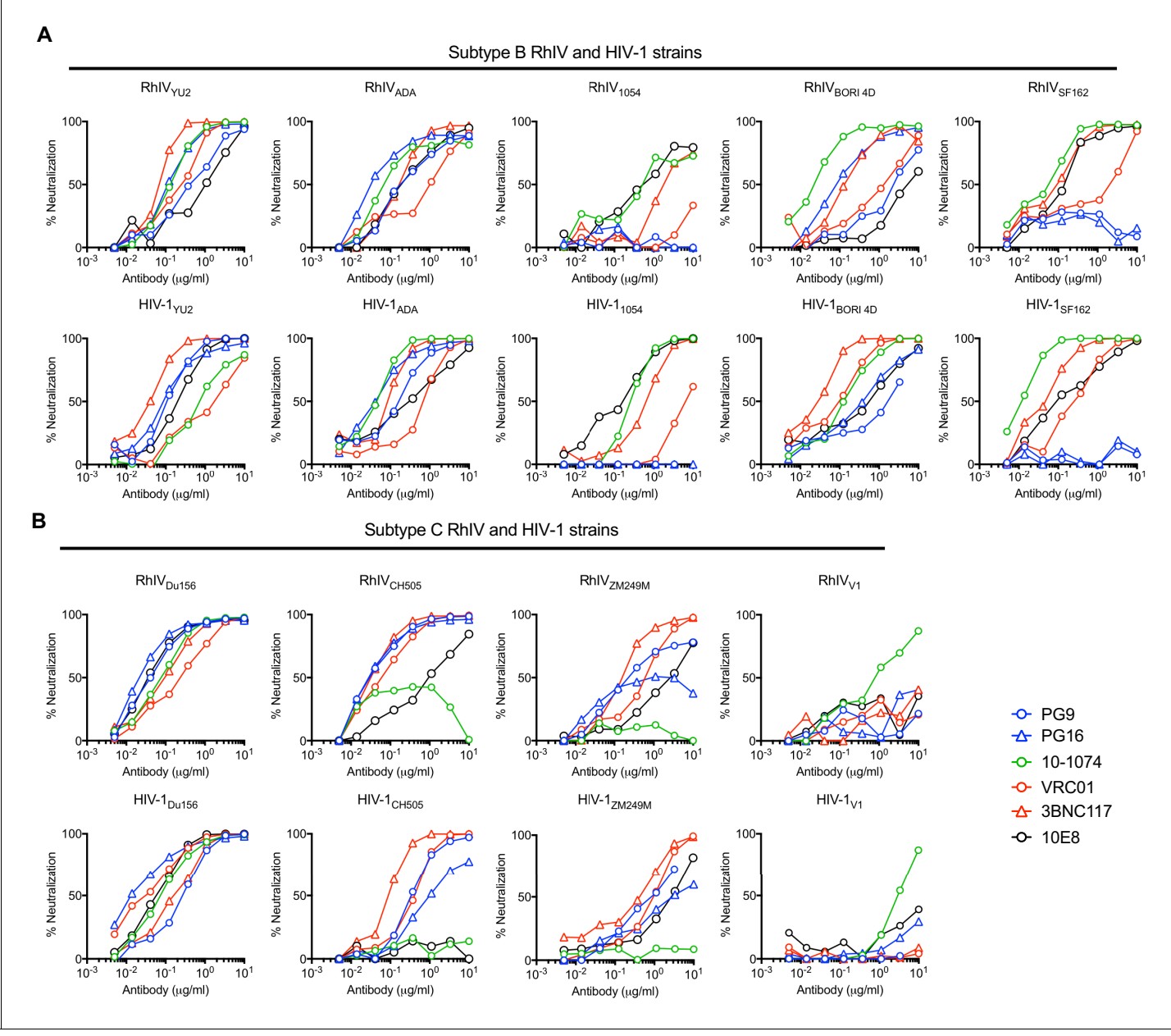

**Figure 2.** Neutralization properties of matched RhIV and HIV-1 strains. (A, B) The indicated RhIV(nLuc) and HIV-1(nLuc) strains bearing subtype B (A) and subtype C (B) envelope proteins were incubated with broadly neutralizing antibodies targeting a V2 quaternary epitope (PG9,PG16), the V3 loop (10–1074), the CD4 binding site (VRC01, 3BNC17) or the MPER (10E8), prior to infection of TZM-Bl cells.

tissue location in humans (see discussion). FACS analysis revealed that hCD4, like mouse CD4, was expressed exclusively on CD3+ cells, but was absent from the CD8+ cell fraction (*Figure 3E*). Overall, 100% of mouse CD4+ cells (but no other cells) in A1 mice expressed hCD4 and CCR5 at levels mimicking human CD4+ T-cells (*Figure 3E*).

## Acute pathology in hCD4/CCR5 transgenic mice following RhIV infection

Because VSV is extremely sensitive to type-1 interferon (*Müller et al., 1994*), we crossed A1, C18 and B4 mice to C57BL/6 mice lacking the type one interferon receptor gene (*Ifnar*1), generating A1$_{Ifnar-/-}$, C18$_{Ifnar-/-}$ and B4$_{Ifnar-/-}$ lines. We first infected A1$_{Ifnar-/-}$ mice with $10^5$ PFU of RhIV$_{CH505}$ by

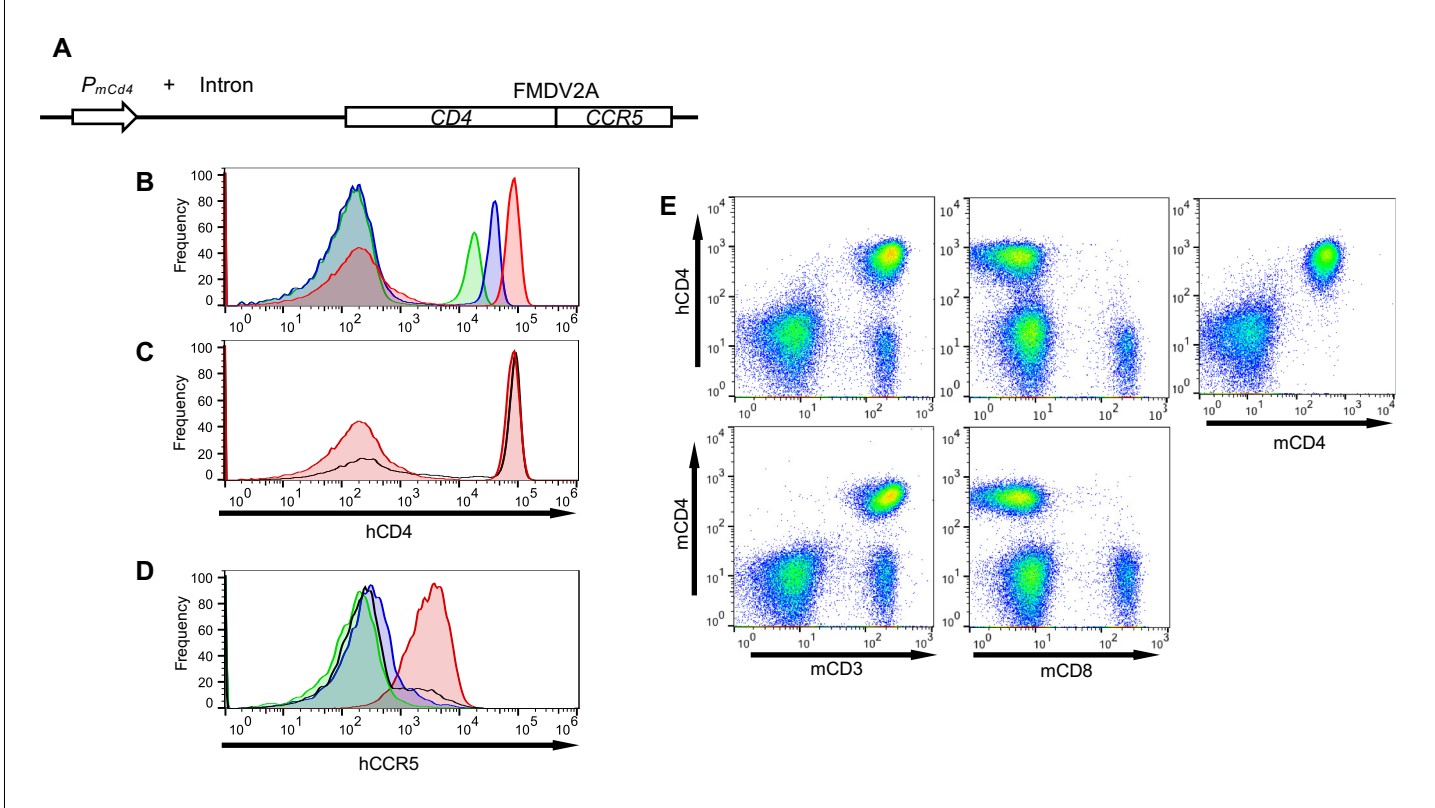

**Figure 3.** Transgenic mice with CD4+ T-cells that express human CD4 and CCR5. (**A**) Schematic representation of the transgene construct that contains a murine *Cd4* promoter and intron 1, linked to human *CD4* and *CCR5* cDNAs separated by sequences encoding an FMDV 2A termination/reinitiation site. (**B**) FACS analysis of hCD4 expression on unfractionated PBMC from three CD4+/CCR5+ transgenic mouse lines: A1 (red histogram) C18 (blue histogram) and B4 (green histogram). (**C**) FACS analysis of hCD4 expression on unfractionated PBMC from transgenic mouse line A1 (red histogram) and a human PBMC donor (black line). (**D**) FACS analysis of CCR5 expression on hCD4+ cells from A1 (red histogram) C18 (blue histogram) and B4 (green histogram) mouse lines and a human PBMC donor (black line). (**E**) FACS analysis of hCD4 expression in combination with mCD3, mCD8 or mCD4.

intraperitoneal injection (i.p.). FACS analysis of peripheral blood mononuclear cells (PBMC) 4 days later revealed profound and selective depletion of CD4+ T-cells (*Figure 4A*). Next, we infected a cohort of A1$_{Ifnar-/-}$ mice with $10^5$ PFU of RhIV$_{CH505}$ and measured CD4+ T-cell numbers and viral RNA levels in lymphoid tissues. This analysis revealed progressive and profound reductions in CD4+ T-cell numbers in PBMC and spleen, and near complete depletion of CD4+ T-cells from thymus and lymph nodes (*Figure 4B*). Viral RNA levels peaked at between $10^3$ and $10^6$ copies/μg of cellular RNA between day 1 and day 4 after infection, depending on the tissue, with the highest levels (>$10^6$ copies/μg) found in thymus (*Figure 4B*).

Conventionally, replication and pathogenesis during immunodeficiency virus infections in primates is monitored longitudinally using blood. We next infected A1$_{Ifnar-/-}$ mice (i.p.) with $10^5$ PFU of RhIV$_{BG505}$, RhIV$_{Du156}$, RhIV$_{SF162}$, or RhIV$_{CH505}$ and monitored plasma viremia and CD4+ T-cells in blood. Plasma viremia peaked at between $10^6$ and $10^8$ RNA copies/ml on day 1 after infection, then declined rapidly during days 1–7 and was cleared before day 14 (*Figure 4C,D*). CD4+ T cells were nearly completely depleted from blood by day 4, then gradually recovered (*Figure 4C,D*). Thus, because analyses of blood enabled long term follow up, and appeared to provide a reasonable surrogate for virus replication and perturbation of cell population in tissues, subsequent analyses were performed using blood.

Infection of A1$_{Ifnar-/-}$ mice with reduced doses of RhIV$_{BG505}$, RhIV$_{Du156}$, RhIV$_{SF162}$, or RhIV$_{CH505}$ revealed that $10^3$ PFU established robust infection with profound CD4+ T-cell depletion, albeit with reduced peak plasma viremia ($10^5$–$10^7$ RNA copies/ml, *Figure 4—figure supplement 1A*). Further reductions in RhIV challenge dose resulted in less consistent infection and reduced peak plasma

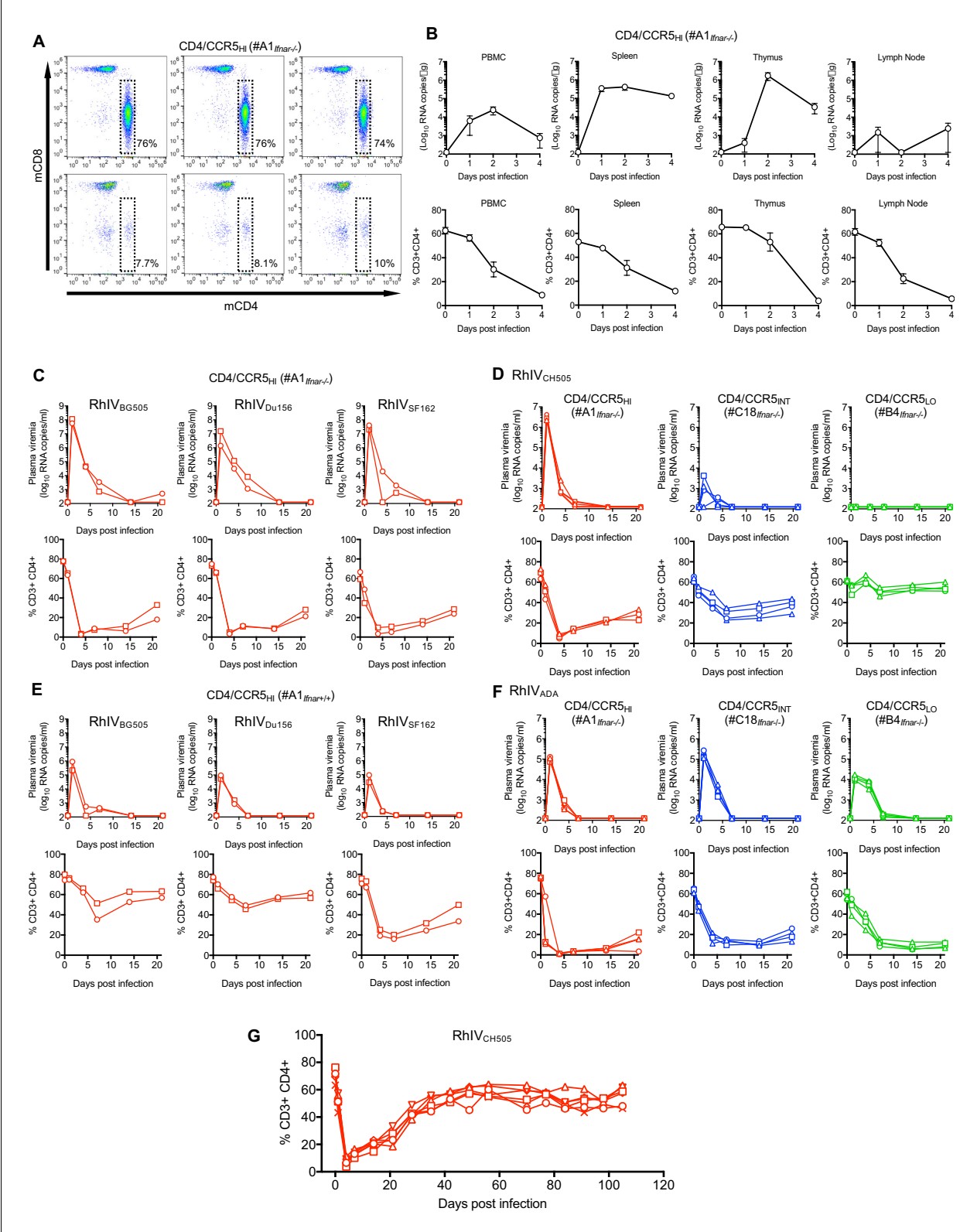

**Figure 4.** RhIV replication and pathology following infection of hCD4/hCCR5 transgenic mice. (A) FACS analysis of CD4 and CD8 expression on T-cells (gated on CD3+ cells) in three A1$_{Ifnar-/-}$ mice prior to RhIV infection (upper row) and 3 days after RhIV$_{CH505}$ infection (lower row). The % of CD3+ cells that were CD4+ is indicated. (B) RhIV RNA levels (log$_{10}$ copies /μg total RNA, upper row) and CD4+ T-cell numbers (% of CD3+ cells, lower row) in A1$_{Ifnar-/-}$ mouse tissues following infection with RhIV$_{CH505}$. Values are the mean ± sd of three mice at each time point. (C–F) RhIV viremia (log$_{10}$ RNA

*Figure 4 continued on next page*

Figure 4 continued

copies/ml of plasma, upper rows) and blood CD4+ T-cell proportion (% of CD3+ cells, lower rows) in A1$_{Ifnar-/-}$ mice (C), A1$_{Ifnar-/-}$, C18$_{Ifnar-/-}$, and B4$_{Ifnar-/-}$ mice (D, F) or A1$_{Ifnar+/+}$ mice (E) at the indicated times following infection with the indicated RhIV strains. Each symbol type on each chart represents an individual mouse (n = 2 to 4 for each virus/mouse strain combination) (G) Blood CD4+ T-cell proportion (% of CD3+ cells) in A1$_{Ifnar-/-}$ mice at the indicated times following infection with RhIV$_{CH505}$. Each symbol type represents an individual mouse (n = 4).
The online version of this article includes the following figure supplement(s) for figure 4:

Figure supplement 1. In vivo RhIV titrations.

viremia, with 0/2, 2/2, 2/2 and 1/2 mice becoming infected with RhIV$_{BG505}$, RhIV$_{Du156}$, RhIV$_{SF162}$, or RhIV$_{CH505}$ at a challenge dose of $10^2$ PFU (*Figure 4—figure supplement 1A*). Even fewer mice became infected with lower peak viremia at a challenge dose of 10 PFU.

Infection of immunocompetent (A1$_{Ifnar+/+}$) mice with $10^5$ PFU of RhIV$_{BG505}$, RhIV$_{Du156}$, or RhIV$_{SF162}$ yielded robust infection albeit with ~10 to 100-fold reduced peak viremia as compared to A1$_{Ifnar-/-}$ mice, (*Figure 4E*). At a lower challenge dose ($10^3$ PFU), A1$_{Ifnar+/+}$ mice gave far less robust viremia than did A1$_{Ifnar-/-}$ mice (*Figure 4—figure supplement 1B*) for all RhIV strains tested. A1$_{Ifnar+/+}$ mice also revealed apparent differences in the ability of RhIV strains to cause CD4+ T-cell depletion that did not correlate with differences in plasma viremia. RhIV$_{SF162}$ appeared to cause more profound CD4+ T-cell depletion than RhIV$_{BG505}$ and RhIV$_{Du156}$, despite reaching similar or lower levels of plasma viremia (*Figure 4E*, *Figure 4—figure supplement 1B*).

We also challenged mice expressing high (A1$_{Ifnar-/-}$), intermediate (C18$_{Ifnar-/-}$) or low (B4$_{Ifnar-/-}$) levels of CD4. The RhIV$_{CH505}$ strain, bearing a T/F HIV-1 Env protein, exhibited exquisite sensitivity to CD4 expression levels. Infection of C18$_{Ifnar-/-}$ mice resulted in 1000-fold lower peak viremia compared to A1$_{Ifnar-/-}$ mice, while B4$_{Ifnar-/-}$ mice appeared completely resistant to RhIV$_{CH505}$ infection (*Figure 4D*). Conversely, the RhIV$_{ADA}$ strain that bears a macrophage tropic HIV-1 Env protein was comparatively insensitive to variation in CD4 expression levels. Indeed, infection of A1$_{Ifnar-/-}$ and C18$_{Ifnar-/-}$ mice gave approximately equivalent peak viremia, while infection of B4$_{Ifnar-/-}$ mice gave peak viremia that was reduced only ~10 fold compared to the other mouse lines (*Figure 4F*).

Although infection of A1$_{Ifnar-/-}$ mice, and in some cases A1$_{Ifnar+/+}$ mice, led to high level viremia and profound CD4+ T-cell depletion, in all cases RhIV infection was apparently cleared. Long term follow-up of a group of RhIV$_{CH505}$-infected A1$_{Ifnar-/-}$ mice showed that CD4+ T-cells exhibited near complete recovery by approximately 40 to 50 days after initial infection (*Figure 4G*).

## Protection of mice against RhIV infection by HIV-1-specific bNAbs

To test the utility of the RhIV model system in evaluating the protective efficacy of antibodies, we challenged mice with RhIV following administration of bNAbs. In the first experiment, A1$_{Ifnar+/+}$ mice were injected subcutaneously (s.c.) with 1 mg of the bNAbs PG16 or 3BNC117 and challenged the following day with RhIV$_{BG505}$. On the basis of previous experiments, this injection is expected to yield 10–100 µg/ml of antibody in the blood of mice (*Klein et al., 2012*). Both antibodies appeared to provide sterilizing protection, in that no viral RNA was detected in plasma of antibody-injected mice and no perturbations in CD4+ T-cells were observed (*Figure 5A*). In a second experiment, A1$_{Ifnar-/-}$ mice were injected s.c. with increasing doses (50 µg −1 mg) of 3BNC117 and challenged the following day with RhIV$_{CH505}$. As was the case with RhIV$_{BG505}$, the highest dose of 3BNC117 gave apparently sterilizing protection against RhIV$_{CH505}$ infection, with undetectable plasma viremia and no CD4+ cell depletion (*Figure 5B*). At 0.5 mg 3BNC117, 2/3 mice exhibited apparently sterile protection, while a third had barely detectable viremia and minimal CD4+ T-cell depletion. At lower 3BNC117 doses (50 µg and 100 µg), partial protection was observed, with low-level viremia ($<10^3$ copies/ml) and clear CD4+ T-cell depletion, that was not as extensive as control animals (*Figure 5B*). In one animal (at the 50 µg dose) apparently complete protection was observed.

## RhIV infection and clearance confers protection against re-infection

The finding that mice cleared RhIV infection provided the opportunity to examine whether protective immune responses might occur following RhIV infection and clearance. Therefore we conducted a series of experiments (Expt #1 through Expt #5), in which mice were challenged three times, several weeks apart, with a single RhIV strain or different RhIV strains (see Materials and methods).

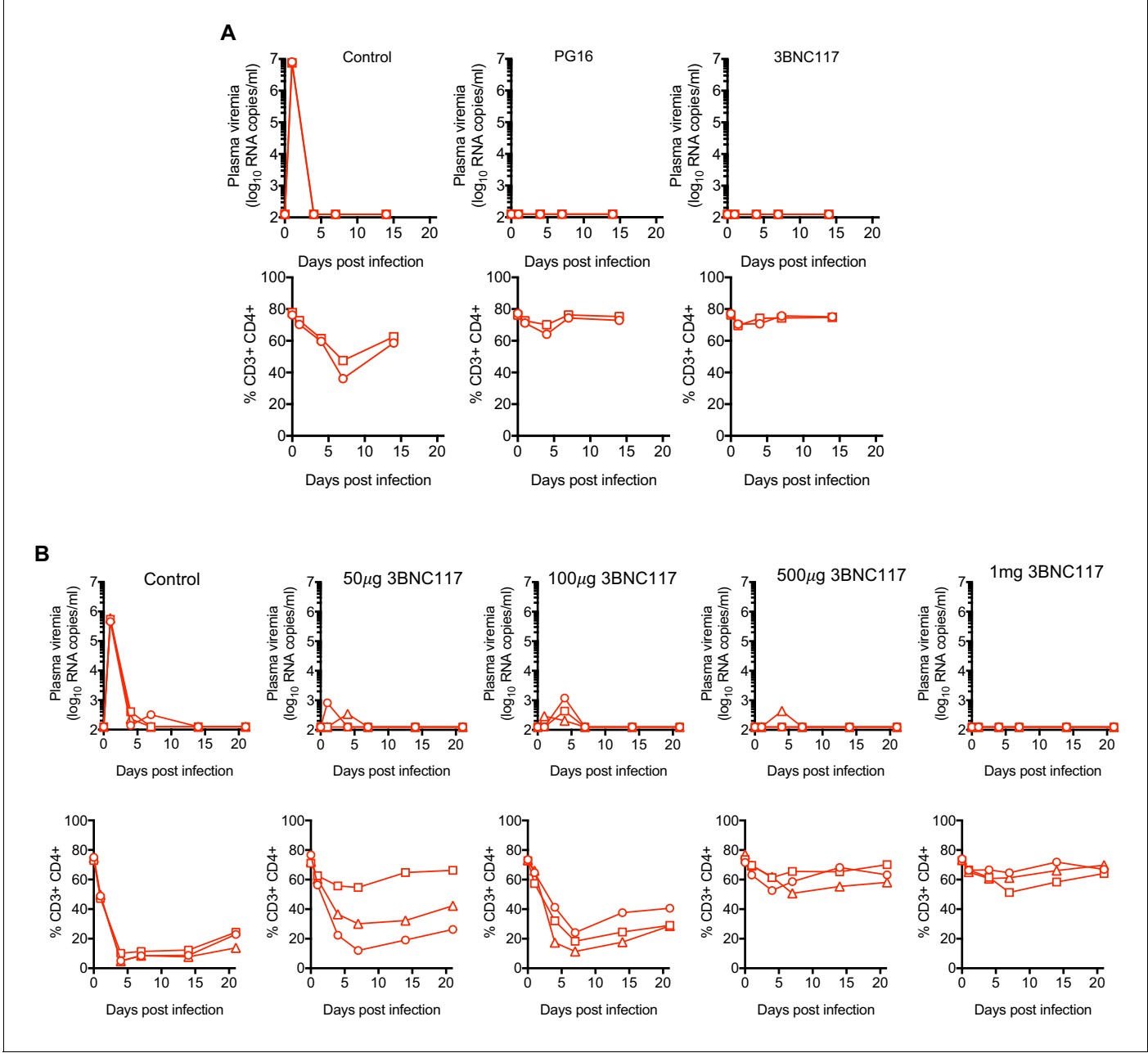

**Figure 5.** Protection against RhIV infection by bNAbs. (A, B) RhIV viremia ($\log_{10}$ RNA copies/ml of plasma, upper rows) and blood CD4+ T-cell proportion (% of CD3+ cells, lower rows) in A1$_{Ifnar+/+}$ mice (A) or A1$_{Ifnar-/-}$ (B) at the indicated times following infection with RhIV$_{BG505}$ (A) or RhIV$_{CH505}$ (B). At 24 hr prior to infection mice were injected (s.c.) with PBS (control) or 1 mg of PG16 or 3BNC117 antibodies (A) or the indicated dose of 3BNC117 antibody (B). Each symbol type represents an individual mouse (n = 2 (A) or n = 3 (B) for each virus/antibody combination).

First, we infected A1$_{Ifnar+/+}$ mice (Expt #1, n = 2, *Figure 6—figure supplement 1A*) or A1$_{Ifnar-/-}$ mice (Expt #2, n = 2, *Figure 6—figure supplement 1B* and Expt #3 n=4 *Figure 6A*) with RhIV$_{SF162}$. As before, RhIV$_{SF162}$ infection was cleared and CD4+ T-cells recovered. At 42 days (Expt #2) or 49 days (Expt#1 and Expt#3) after the first infection, mice were rechallenged with RhIV$_{SF162}$. Following the second challenge, only low-level plasma viremia (~$10^3$ RNA copies/ml) was detected, and only in a subset of mice. The magnitude of CD4+ T-cell depletion following the second infection was reduced compared to the first infection (in A1$_{Ifnar-/-}$ mice) or absent (in A1$_{Ifnar+/+}$ mice). Following a third challenge with RhIV$_{SF162}$ at 84 days (Expt#2) or 91 days (Expt#1 and Expt#3) after the first

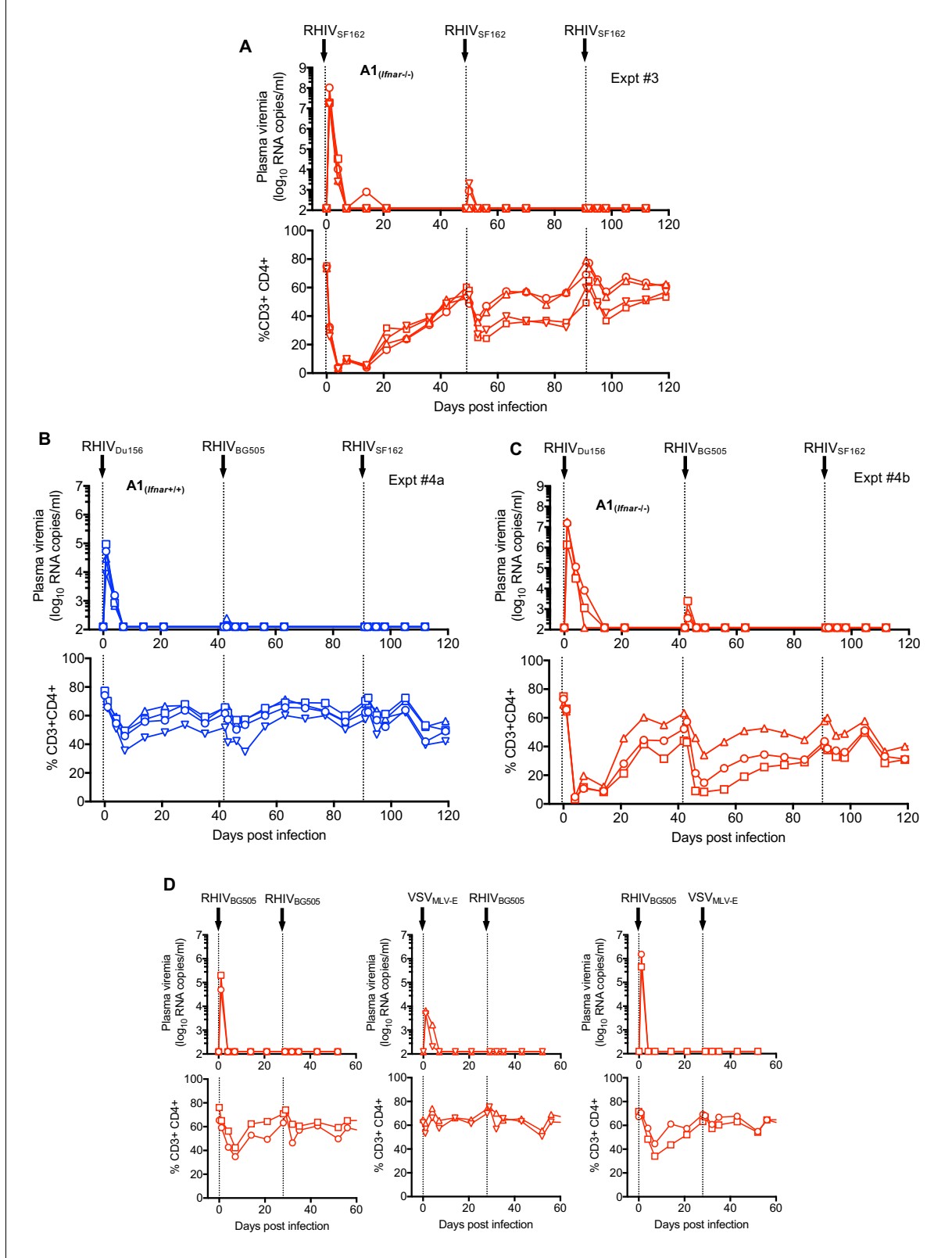

**Figure 6.** Protection against RhIV reinfection. (**A**) RhIV viremia (log₁₀ RNA copies/ml of plasma, upper rows) and blood CD4+ T-cell proportion (% of CD3+ cells, lower rows) in A1$_{Ifnar-/-}$ mice following infection with RhIV$_{SF162}$ on days 0, 49 and 91. Each symbol type represents an individual mouse (n = 4). (**B, C**) RhIV viremia (log₁₀ RNA copies/ml of plasma, upper rows) and blood CD4+ T-cell proportion (% of CD3+ cells, lower rows) in A1$_{Ifnar+/+}$ mice (**B**) and A1$_{Ifnar-/-}$ mice (**C**) following infection with RhIV$_{Du156}$, RhIV$_{BG505}$, and RhIV$_{SF162}$ on days, 0, 42 and 91, respectively. Each symbol type

*Figure 6 continued on next page*

Figure 6 continued

represents an individual mouse (n = 4). (D) RhIV$_{BG505}$ and VSV$_{MLV-E}$ viremia (log$_{10}$ RNA copies/ml of plasma, upper rows) and blood CD4+ T-cell proportion (% of CD3+ cells, lower rows) in A1$_{Ifnar-/-}$ mice following infection with RhIV$_{BG505}$ on day 0 and day 28 (left panels), VSV$_{MLV-E}$ on day 0 and RhIV$_{BG505}$ day 28 (center panels) or RhIV$_{BG505}$ on day 0 and VSV$_{MLV-E}$ day 28 (right panels).

The online version of this article includes the following figure supplement(s) for figure 6:

**Figure supplement 1.** Repeat infections of mice with homologous or heterologous RhIV strains.
**Figure supplement 2.** Construction of VSV$_{MLV-E}$ and Vaccine effect of RhIV infection does not require B-cells.
**Figure supplement 3.** Gp160 (SOSIP) binding antibodies in RhIV infected mice (Expt #3 and #4).
**Figure supplement 4.** Gp160 (SOSIP) binding antibodies in RhIV infected mice (Expt #5).
**Figure supplement 5.** Gp160 (SOSIP) binding antibodies in RhIV infected mice (Expt #5).
**Figure supplement 6.** Characterization of antibodies in RhIV infected mice.

infection, plasma viremia was undetectable, and only minor perturbations of CD4+T cell numbers were observed (*Figure 6A*, *Figure 6—figure supplement 1A,B*).

Next, we did similar experiments (Expt #4a and Expt #4b) that employed three sequential challenges with RhIV strains bearing different HIV-1 Env subtypes at each challenge. First, we infected A1$_{Ifnar+/+}$ mice (Expt #4a, n = 4, *Figure 6B*) and A1$_{Ifnar-/-}$ mice (Expt #4b, n = 4, *Figure 6C*) with RhIV$_{Du156}$ (encoding a subtype C HIV-1 Env). As expected, plasma viremia was cleared within 1 to 2 weeks and CD4+T cells were depleted but then recovered. At 42 days and 91 days after the initial RhIV$_{Du156}$ infection, mice were challenged with RhIV$_{BG505}$ (subtype A Env) and RhIV$_{SF162}$ (subtype B Env), respectively. The second challenge with RhIV$_{BG505}$ resulted in only low-level plasma viremia in a subset of mice and an attenuated degree of CD4+ T-cell depletion. The third challenge with RhIV$_{SF162}$ gave no detectable plasma viremia and minimal CD4+ T-cell depletion (*Figure 6B,C*). Similar results were obtained in Expt #5, where mice were infected on three occasions with various combinations of homologous or heterologous RhIV strains with envelopes of various subtypes (*Figure 6—figure supplement 1C,D,E and F*). Overall, infection with RhIV gave an apparent 'vaccine' effect, that is there was immunity to subsequent challenge with homologous or heterologous RhIV strains, that exhibited breadth with respect to the HIV-1 envelope protein encoded by the challenge strain.

## Protection against RhIV re-infection does not require B-cells or a homologous Env protein

To begin to ascertain whether HIV-1 Env-specific antibodies contributed to the apparent vaccine effect of initial RhIV infections on subsequent RhIV challenges, we constructed another VSV-derived chimeric virus, termed VSV$_{MLV-E}$ (GFP). The design of VSV$_{MLV-E}$ (GFP) was the same as RhIV, except that it encoded an Env ectodomain and transmembrane sequences from ecotropic murine leukemia virus (MLV-E) rather than HIV-1 (*Figure 6—figure supplement 2A*). VSV$_{MLV-E}$ was also equipped with a *GFP* reporter gene, and replicated well in NIH3T3 cells (*Figure 6—figure supplement 2B*), yielding cell-free titers of ~10$^6$ PFU/ml.

We challenged A1$_{Ifnar+/+}$ mice with VSV$_{MLV-E}$ or RhIV$_{BG505}$ which resulted in transient plasma viremia of ~10$^4$ RNA copies/ml (VSV$_{MLV-E}$) or 10$^5$ to 10$^6$ RNA copies/ml (RhIV$_{BG505}$) (*Figure 6D*). A subsequent challenge, 28 days later, with RhIV$_{BG505}$ resulted in undetectable plasma viremia, and little or no CD4+ T-cell depletion, whether mice had been previously infected with VSV$_{MLV-E}$ or RhIV$_{BG505}$ (*Figure 6D*). Similarly, challenge of A1$_{Ifnar+/+}$ mice with VSV$_{MLV-E}$, 28 days after infection and clearance of RhIV$_{BG505}$ resulted in no detectable plasma viremia (*Figure 6D*). Given that MLV-E and HIV-1 Env proteins share no sequence similarity, these experiments suggested that the protection against RhIV infection, afforded by a prior RhIV infection (*Figure 6* and *Figure 6—figure supplement 1*) did not require an immune response to the HIV-1 envelope protein.

To further explore whether antibody responses might be responsible for the vaccine effect of RhIV infection, we crossed A1$_{Ifnar+/+}$ mice to µMT-/-mice that lack functional B-cells (*Kitamura et al., 1991*). Then, A1$_{Ifnar+/+}$ and A1$_{Ifnar+/+,\ µMT-/-}$ mice were challenged with RhIV$_{CH505}$. Similar and characteristic trajectories of RhIV$_{CH505}$ plasma viremia and transient CD4+ T-cell depletion were observed in both B-cell competent and B-cell deficient mouse strains (*Figure 6—figure supplement 2C*). Following rechallenge with RhIV$_{CH505}$ 48 days later, neither A1$_{Ifnar+/+}$ nor A1$_{Ifnar+/+,\ µMT-/-}$ mice exhibited plasma viremia or CD4+T-cell depletion (*Figure 6—figure supplement 2C*). Thus, a B-cell mediated

immune response was not required for the vaccine effect of a prior RhIV infection on subsequent RhIV challenge.

## Serological responses to HIV-1 env in RhIV-infected mice

Although the above experiments indicated that antibodies were not essential for protection from a secondary RhIV challenge, they did not determine whether or not protective antibodies might be present. We therefore collected plasma from mice that had been repeatedly challenged with RhIV strains in Expt #1 to Expt #5 (*Figure 6A–C* and *Figure 6—figure supplement 1A–F*) and tested for the presence of antibodies capable of Env binding, neutralization and protection.

Antibody binding tests employed subtype A, B and C SOSIP Env proteins (*Sanders et al., 2013*), captured at their C-termini on ELISA plates (*Figure 6—figure supplements 3–5*). A1$_{Ifnar-/-}$ mice infected three times with RhIV$_{SF162}$ (subtype B, Expt #3) elicited antibodies that bound all four of the SOSIP envelope proteins, whose titers increased after the second infection (*Figure 6—figure supplement 3A*). ELISA titers were higher with the B41 (subtype B) and BG505 (subtype A) than with the two subtype C SOSIP proteins (*Figure 6—figure supplement 3A*), partly reflecting sequence similarity between the infecting RhIV strain and the ELISA antigens. Mice infected sequentially with RhIV$_{Du156}$, (subtype C), RhIV$_{BG505}$ (subtype A) then RhIV$_{SF162}$ (subtype B), in Expt #4 generated antibodies with higher titers on two subtype C SOSIP proteins and the BG505 SOSIP protein than the B41 (subtype B SOSIP) protein (*Figure 6—figure supplement 3B*). However, all four SOSIP proteins were recognized and the second infection (RhIV$_{BG505}$) boosted and apparently broadened the antibody response initiated by RhIV$_{Du156}$ infection. A1$_{Ifnar-/-}$ mice generated higher titers of Env binding antibodies than A1$_{Ifnar+/+}$ mice, thus any potential deficit in antibody generation that might have resulted from the absence of type-I interferon signals was overwhelmed by the larger antigen load following infection (*Figure 6—figure supplement 3B*). Antibody titers following sequential infection with various combinations of RhIV strains in Exp #5 (*Figure 6—figure supplement 4A and B*, *Figure 6—figure supplement 5A,B*) followed a general pattern that the antigen most closely resembling the initial challenge virus was best recognized in ELISA assays conducted after the first infection, although the BG505 SOSIP appeared to be generally better recognized than the other SOSIP proteins. The second and sometimes the third RhIV challenges with either homologous or heterologous RhIV strains increased titers and broadened ELISA reactivity (*Figure 6—figure supplement 4A and B*, *Figure 6—figure supplement 5A,B*). In most cases, broadly Env reactive binding antibodies were elicited after three RhIV infections, regardless of the RhIV stains used in the three challenges.

Further analysis of sera from a subset of mice challenged with a variety of RhIV strains (in Exp #5) using commercial antigen-loaded diagnostic reagents (INNO-LIA HIV I/II Score) revealed prominent reactivity with epitope(s) on gp41 as well as gp120 (*Figure 6—figure supplement 6A*). Notably, however, these mouse sera did not contain detectable levels of antibodies that could compete with human bnAbs or (sCD4) for binding to the CD4bs, the apex of the Env trimer formed by the V2 loop or a glycosylation dependent epitope in the V3 loop (*Figure 6—figure supplement 6B*).

We next tested neutralization activity of immunoglobulins purified from pooled convalescent sera taken from mice after the three sequential RhIV challenges. In mice that had been challenged three times with RhIV$_{SF162}$, (Expts #1–3) weak neutralization activity was observed against HIV-1$_{SF162}$, but not against a heterologous strain (HIV-1$_{CH505}$) (*Figure 7A*). In mice that were sequentially infected three times with RhIV$_{Du156}$, RhIV$_{BG505}$ and RhIV$_{SF162}$, (Expt #4) no neutralization was detected against HIV-1$_{Du156}$, HIV-1$_{BG505}$ or HIV-1$_{SF162}$ (*Figure 7B*). Similarly, in mice that were sequentially infected three times with various RhIV strains (Exp #5), no neutralization was observed against any of the matched HIV-1 strains (*Figure 7C*). Even three challenges with RhIV$_{BG505}$ failed to elicit neutralizing activity against HIV-1$_{BG505}$. This finding contrasts with the results obtained with RhIV$_{SF162}$/HIV-1$_{SF162}$. Overall, RhIV infection elicited high titers of HIV-1 envelope binding antibodies. However, these antibodies were primarily non-neutralizing.

## Partial protection conferred by passive transfer of RhIV-convalescent sera

Although most sera from infected mice lacked HIV-1 neutralization activity, it was possible that non-neutralizing antibodies might contribute to protection (e.g. via antibody-dependent cellular

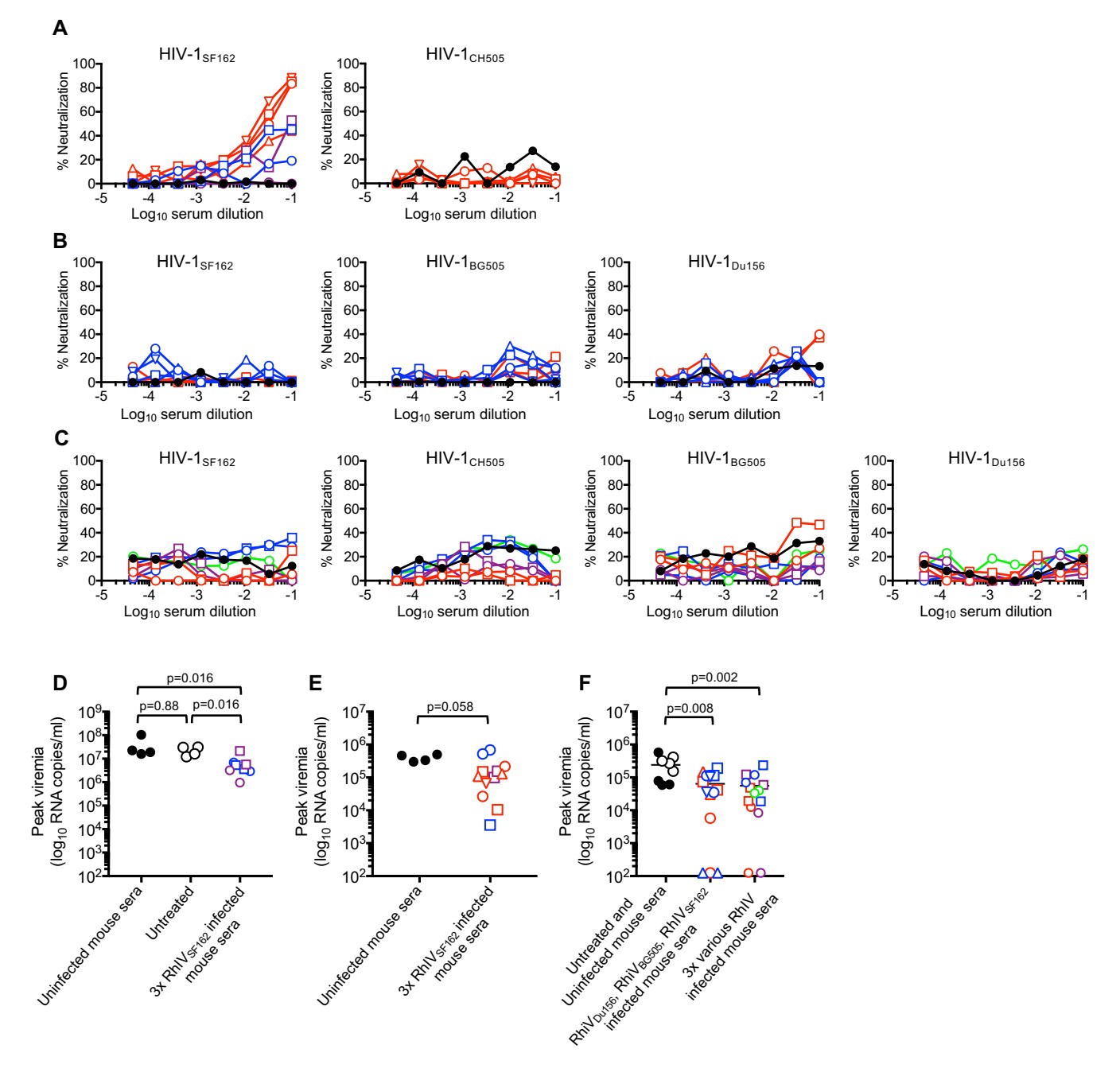

**Figure 7.** Neutralization and passive protection activity in sera from RhIV infected mice. (**A**) Neutralization (using TZM-Bl target cells) of HIV-1(nLuc) strains by immunoglobulins purified from convalescent mouse sera after three infections with RhIV$_{SF162}$. Symbol types and line colors correspond to individual mice in Expts #1 to #3 (depicted in *Figure 6A*, *Figure 6—figure supplement 1A and B*). Black symbols indicate sera from uninfected mice (**B**) Neutralization of HIV-1(nLuc) strains by immunoglobulins purified from convalescent mouse sera after infection with RhIV$_{Du156}$, RhIV$_{BG505}$ and RhIV$_{SF162}$ in Expt #4. Symbol types and line colors correspond to individual mice (depicted in *Figure 6B and C*). (**C**) Neutralization (using TZM-Bl target cells) of HIV-1(nLuc) strains by immunoglobulins purified from convalescent mouse sera after three infections with various RhIV strains in Expt #5. Symbol types and line colors correspond to individual mice depicted in *Figure 6—figure supplement 1C,D,E and F*. (**D**) Peak viremia (day one post infection with $10^5$ PFU RhIV$_{SF162}$) following no treatment or passive administration of sera from uninfected mice, or mice that had previously been infected three times with RhIV$_{SF162}$. Colored symbols indicate different donor mice, matched to correspond to donor mice from Expt #1 and #2 (depicted in *Figure 6—figure supplement 1A and B*). Black closed circles indicate sera from uninfected mice, Black open circles indicate no serum treatment. (**E**) Peak viremia (day one post infection with $10^3$ PFU RhIV$_{SF162}$) following passive administration of sera from uninfected mice, or mice that had previously been infected three times with RhIV$_{SF162}$. Colored symbols indicate different donor mice, matched to correspond to donor mice from

*Figure 7 continued on next page*

*Figure 7 continued*

Expt #1 - #3 (depicted in *Figure 6A*, *Figure 6—figure supplement 1A and B*). (F) Peak viremia (day one post infection with $10^3$ PFU RhIV$_{SF162}$) following no treatment, passive administration of sera from uninfected mice, or mice that had previously been infected with RhIV$_{Du156}$, RhIV$_{BG505}$ and RhIV$_{SF162}$ (Expt #4). Alternatively, serum from mice sequentially infected with various combinations of three RhIV strains (Expt #5) were used. Colored symbols indicate different donor mice, matched to correspond to donor mice from Expt #4 or Expt #5 (depicted in *Figure 6B and C*, or *Figure 6—figure supplement 1C,D,E and F*).

The online version of this article includes the following figure supplement(s) for figure 7:

**Figure supplement 1.** Passive serum transfer/protection experiments using donor serum from RhIV infected mice.

cytotoxicity, ADCC). Therefore, we collected sera from mice after three challenges in Expts #1 to #5 and conducted passive protection experiments. Convalescent serum from each infected mouse was injected s.c. into two recipient mice that were challenged i.p. the following day with RhIV$_{SF162}$. First, recipients were given serum from Expt #1 - #3 donors that had been infected three times with RhIV$_{SF162}$ and had weak neutralizing activity that was specific to HIV-1$_{SF162}$ (*Figure 7A*). Then, recipients were challenged with either $10^5$ PFU (*Figure 7D*) or $10^3$ PFU (*Figure 7E*) RhIV$_{SF162}$. Reduced peak plasma viremia was observed in mice that had received convalescent serum compared to controls (*Figure 7D,E*, *Figure 7—figure supplement 1A,B,C*). However, none of the recipient mice were completely protected, and the reduction in plasma viremia was not statistically significant in one of the recipient mouse cohorts (*Figure 7E*). Serum from mice that had been infected with RhIV$_{Du156}$, RhIV$_{BG505}$, and RhIV$_{SF162}$, (Expt #4, *Figure 6B and C*) lacked neutralizing activity but was nevertheless weakly protective. Indeed, upon challenge with $10^3$ PFU RhIV$_{SF162}$, Expt #4 convalescent serum recipients had lower peak viremia than controls, and three of fourteen mice were completely protected, with no detectable viremia and no depletion of CD4+ T-cells (*Figure 7F*, *Figure 7—figure supplement 1D*). Serum from mice that had been infected with various combinations of 3 RhIV strains (Expt #5) also lacked neutralizing activity and was also weakly protective. Upon challenge with $10^3$ PFU RhIV$_{SF162}$, convalescent serum recipients again had lower peak viremia than controls, and two out of fourteen mice were completely protected (*Figure 7F*, *Figure 7—figure supplement 1E*). Overall, convalescent serum from RhIV infected mice had abundant and broad Env binding activity, and weak protective activity in passive transfer experiments, that that did not correlate with the presence or absence of neutralizing antibodies.

## Discussion

Herein, we describe the development and use of a small animal, chimeric virus-challenge model that captures several key features of HIV-1 infection. Specifically, RhIV strains exhibit the tropism and, to a large extent, the neutralization properties of HIV-1. The transgenic mice that are a key component of the RhIV model expressed hCD4 and hCCR5 at the same level as human T-cells, and this expression was restricted to T-cells that normally express mCD4. RhIV infection results in acute, resolving viremia and CD4+ T-cell depletion. This model can therefore be used to test the activity of monoclonal and polyclonal antibodies in an in vivo setting, where the effector functions of antibodies can contribute to their antiviral activity in a way that is difficult to recapitulate in vitro.

Because RhIV employs the intracellular replication machinery of VSV, there are some obvious caveats associated with the RhIV/mouse model. First, the replication cycle of VSV is more rapid than that of HIV-1, with a single cycle of replication typically requiring 6–8 hr (*Cuevas et al., 2005*). Second, the mode of replication ('stamping machine' with a DNA provirus for HIV-1, versus geometric RNA replication for VSV) might affect the propensity to accumulate escape mutations under antibody driven selective pressure (*Safari and Roossinck, 2014*). In practice, however, the early onset of peak viremia and rapid clearance of RhIV infection precluded an assessment of RhIV evolution in the presence and absence of selective pressure. The absence of latency in VSV replication is also a key distinction from HIV-1 infection. The mice generated herein differ from humans in that 100% of their CD4+T-cells also expressed CCR5, because of the tight linkage of CD4 and CCR5 in the transgene construct. In humans, the proportion of CD4+ T-cells that also express CCR5 varies according to tissue source and inflamation; approximately 5–10% of peripheral blood CD4+ T-cells express CCR5, while most gut associated lymphoid tissue and rheumatoid arthritis synovial fluid CD4+ T-cells are

CCR5+ (*Agace et al., 2000*; *Qin et al., 1998*). The elevated frequency with which CCR5 is expressed in blood cells in our transgenic mice may accelerate CD4+ T-cell depletion during RhIV infection.

A property of RhIV particles that might be relevant to tropism and neutralization properties is spike density (the number of envelope trimers per virion). HIV-1 has a lower spike density than does VSV (*McSharry et al., 1971*; *Zhu et al., 2006*). While we were unable to precisely determine the spike density on RhIV particles, our semi-quantitative estimates indicated that the number of envelope proteins on a RhIV particle was greater than that present on an HIV-1 particle but less than that on a VSV particle. Importantly, however, RhIV strains mimicked the diversity of properties associated with parental HIV-1 strains. A RhIV strain generated using a T/F envelope protein (RhIV$_{CH505}$) exhibited a requirement, typical of that associated with many T/F HIV-1 strains, for the high hCD4 levels found on human T-cells (*Chikere et al., 2014*). Conversely, a RhIV strain constructed using a macrophage tropic envelope (RhIV$_{ADA}$) replicated well in mice whose T-cells expressed lower levels of hCD4 (*Joseph et al., 2014*). Most crucially, the neutralization properties of RhIV strains were similar to those of HIV-1 strains bearing a cognate envelope protein.

An interesting feature of RhIV infection was that mice seroconverted to the HIV-1 Env proteins and generated a prominent Env binding antibody response, albeit one that was largely non-neutralizing. The low titer neutralizing antibodies that were elicited by repeated RhIV$_{SF162}$ infection were autologous, and strain specific neutralization was evident only against an easy-to-neutralize HIV-1 strain, SF162 (*Seaman et al., 2010*). The relatively poor generation of neutralizing antibodies should be expected, given the short period of viremia associated with RhIV infection. In HIV-1 infected humans, autologous neutralizing antibodies typically arise after months of persistent HIV-1 infection, with neutralization breadth only developing (to varying degrees) over the ensuing years of chronic infection (*Landais and Moore, 2018*). Elaborations of this model that would increase the amount of time that replication occurs in the presence of neutralizing antibodies (for example substitution of HIV-1 Env into other viruses that are capable of replication in mice, or ablation of CD8-mediated cytotoxic responses) may allow the co-evolution of Env sequence and antibody to be studied.

Additionally, the apparently unfavorable nature of the mouse immunoglobulin repertoire may also contribute to the absence of neutralizing antibodies in the current iteration of the RhIV model, as C57BL/6 mice have previously been reported to be unable to generate autologous neutralizing antibodies following BG505 SOSIP protein immunization (*Hu et al., 2015*). Future iterations of the RhIV model could benefit from the crossbreeding of A1 mice to mouse strains with human immunoglobulin repertoires (*Lee et al., 2014*; *Murphy et al., 2014*), although the availability of the such mouse strains to the academic community remains restricted.

Despite the paucity of neutralizing antibodies generated by RhIV infected mice, primary infection conferred at least partial protection against a second, and especially a third, RhIV challenge. The bulk of this 'vaccine' effect was Env-independent and likely cell mediated. This finding suggests that protective immunity against a cytopathic, CD4+ T-cell tropic virus can, at least in principle, be established without protective antibodies. Nevertheless, convalescent serum from animals that had been challenged three times with RhIV strains exhibited modest protective efficacy in passive transfer experiments with an RhIV$_{SF162}$ challenge. This protective activity was evident even in the absence of in vitro neutralizing activity. It is therefore likely that protection was mediated by an effector-dependent activity of the antibodies present therein, although we cannot exclude the possibility that sub-detectable neutralizing activity might be responsible. The protection afforded by convalescent sera was limited and manifested as modest reductions in acute RhIV$_{SF162}$ viremia in most mice. In other animal models, and perhaps in the context of human vaccination (*Rerks-Ngarm et al., 2009*), non-neutralizing antibodies against HIV-1 Env may also be responsible for modest protection (*Haynes et al., 2012*). Conversely, we found that broadly neutralizing antibodies readily conferred apparently sterilizing protection upon challenge with RhIV$_{CH505}$ and RhIV$_{BG505}$ that bear Env proteins from more representative HIV-1 strains. The apparent protective effect of non-neutralizing antibodies in this model could, potentially, be enhanced by the absence of accessory genes that mediate CD4 or tetherin downregulation. Entrapment of envelope protein or virions on the surface of infected cells or exposure of CD4-induced epitopes may sensitize infected cells to effector-dependent activities.

In conclusion, we have developed a virus-host model system that recapitulates some key features of acute HIV-1 infection. The genetic manipulability of both host and virus in this model could permit

a wide range of studies on the factors that influence the elicitation of HIV-1 Env specific antibodies and antiviral efficacy of neutralizing and non-neutralizing antibodies and sera in vivo.

# Materials and methods

**Key resources table**

| Reagent type (species) or resource | Designation | Source or reference | Identifiers | Additional information |
|---|---|---|---|---|
| Strain, strain background (Vesicular Stomatitis Virus) | pVSV-FL+(2) Plasmid Expression Vector System | Kerafast | Cat#EH1002 | Anti-genomic sense plasmid with helper plasmids N, P, G and L |
| Genetic reagent (*Mus musculus*) | C57BL/6J-Tg (Cd4-CD4,CCR5)A1Bsz; C57BL/6J-Tg(Cd4-CD4, CCR5)C18Bsz; C57BL/6J-Tg(Cd4-CD4, CCR5)B4Bsz | This paper | | Mouse lines with CD4 cell-specific expression of human CD4 and CCR5 |
| Cell line (*H. sapiens*) | 293T | ATCC | CRL-3216 | |
| Cell line (*H. sapiens*) | GHOSTX4; GHOSTR5 | NIH AIDS Reagent Repository | Cat#3685;3944 | |
| Cell line (*H. sapiens*) | MT2 | NIH AIDS Reagent Repository | Cat#237 | |
| Antibody | Anti-mouse CD16/CD32 (purified rat monoclonal) | BD Pharmingen | Cat#553142 | FACS (2 uL per test) |
| Antibody | FITC Anti-mouse CD3 (rat monoclonal) | BD Pharmingen | Cat#555274 | FACS (2 uL per test) |
| Antibody | PerCP-Cy5.5 Anti-mouse CD4(rat monoclonal) | BD Pharmingen | Cat#550954 | FACS (2 uL per test) |
| Antibody | APC Anti-mouse CD8a (rat monoclonal) | Biolegend | Cat#100712 | FACS (1 uL per test) |
| Antibody | APC-Cy7 Anti-human CD4(mouse monoclonal) | Biolegend | Cat#317418 | FACS (2 uL per test) |
| Antibody | PE Anti-mouse CD19 (rat monoclonal) | BD Pharmingen | Cat#553786 | FACS (1 uL per test) |
| Antibody | PE Anti-human CD195/CCR5 (mouse monoclonal) | BD Pharmingen | Cat#560935 | FACS (2.5 uL per test) |
| Antibody | PE Anti-human CD195 (mouse monoclonal) | BD Pharmingen | Cat#550632 | FACS (2.5 uL per test) |
| Antibody | AlexaFluor 647 Anti-human CD4 (mouse monoclonal) | Biolegend | Cat#300520 | FACS (3 uL per test) |
| Antibody | Anti-HIV-1 gp120 (goat polyclonal) | American Research Products | Cat#12-6205-1 | WB (1:1000) |
| Antibody | Anti-VSV M (mouse monoclonal) | Kerafast | Cat#EB0011 | WB (1:2000) |
| Antibody | His-Tag Antibody (pAb, Rabbit) | GenScript | A00174-40 | ELISA coating at 0.5 mg/ml |
| Antibody | Goat anti-mouse IgG H and L (HRP) preadsorbed | Abcam | Ab97040 | ELISA (1:20000) |
| Antibody | Goat anti-human IgG H and L (HRP) preadsorbed | Abcam | Ab97175 | ELISA (1:20000) |

*Continued on next page*

*Continued*

| Reagent type (species) or resource | Designation | Source or reference | Identifiers | Additional information |
|---|---|---|---|---|
| Recombinant DNA reagent | pLHCX hCD4 2A CCR5 (plasmid) | This paper | | Retroviral vector with human CD4/CCR5 |
| Recombinant DNA reagent | pNL1.1 (plasmid) | Promega | #N1001; GenB:JQ437370 | Nanoluciferase cDNA |
| Recombinant DNA reagent | pAAVCMV_BG505-His | This paper | | plasmid expressing his-tagged BG505 SOSIP |
| Recombinant DNA reagent | pAAVCMV_B41-His | This paper | | plasmid expressing his-tagged B41 SOSIP |
| Recombinant DNA reagent | pAAVCMV_Du422-His | This paper | | plasmid expressing his-tagged Du422 SOSIP |
| Recombinant DNA reagent | pAAVCMV_Zm197-His | This paper | | plasmid expressing his-tagged Zm197 SOSIP |
| Sequence-based reagent | RL413 | This paper | Genotyping PCR primer | GAACCTGGTGGTGAT GAGAGCCACTCA |
| Sequence-based reagent | RL425 | This paper | Genotyping PCR primer | TGCTTGCTTTAACA GAGAGAAGTTCGT |
| Sequence-based reagent | RL509 | PMID: 16617693 | RT-qPCR primer | TGATACAGTACAAT TATTTTGGGAC |
| Sequence-based reagent | RL510 | PMID: 16617693 | RT-qPCR primer | GAGACTTTCTGTT ACGGGATCTGG |
| Chemical compound, drug | Maraviroc | NIH AIDS Reagent Repository | Cat#11580 | |

## Rhabdo immunodeficiency virus (RhIV) clones

Plasmids encoding the full length VSV genome (pVSV-FL) as well as individual VSV genes N, P, L, and G were purchased from Kerafast (VSV-FL+[2] VSV Plasmid Expression Vector System, EH1002). Plasmids encoding individual HIV-1 *env* genes were obtained from the NIH AIDS regent repository. Alternatively, *env* sequences were synthesized (Genart, Thermofisher). Chimeric envelope genes were generated using overlapping PCR products, in which the ectodomain and transmembrane domains of each HIV-1 Env (equivalent to HIV-1 $_{HXB2}$ amino acids 1–709) was fused to the cytoplasmic tail of VSV-G (amino acids 486–511, *Figure 1A*). The chimeric Env cDNAs were inserted into pVSV-FL precisely in place of the existing VSV-G encoding sequences to generate pRhIV plasmids encoding chimeric HIV-1/VSV-G envelopes. VSV$_{MLV-E}$ had a similar design, except that MLV-E Env ectodomain and transmembrane domains (amino acids 1–634) were fused to the cytoplasmic tail of VSV-G (amino acids 486–511, see *Figure 6—figure supplement 2A*).

RhIV viruses were generated by infecting 293 T cells with T7-expressing vaccinia (vTF7-3) at a MOI of 5, followed by transfection with pRhIV plasmids and plasmids encoding VSV-N, P, L, and G under the control of a T7 promoter. Supernatants were harvested 48 hr post transfection, filtered (0.2 µm) to remove the bulk of the vaccinia virus and plaque purified on GHOST R5 cells. Plaque purified virus was expanded on 293T CD4/R5 cells and cell culture supernatant was harvested, passed through a 0.2 µm filter and frozen in aliquots. Virus titers (PFU/ml) were determined by plaque formation using GHOST R5 cells. For in vitro spreading replication assays (*Figure 1*), GHOST R5 cells were infected with RhIV stocks MOI of $10^{-4}$. Thereafter, aliquots of culture supernatants were harvested at the indicated times 15–40 hr after infection and the extracellular virus yield determined by titration and plaque assay on GHOST R5 cells.

RhIV derivatives encoding nano luciferase (nLuc) were generated by inserting the nLuc encoding sequences (from pNL1.1, Promega) into pRhIV plasmids between the envelope and L genes, along with appropriate VSV regulatory sequences. A pRhIV plasmid encoding GFP was similarly generated

by inserting the EGFP encoding sequences between the envelope and L genes, along with appropriate VSV regulatory sequences.

## HIV-1 reporter viruses

HIV-1 proviral plasmids expressing various Env genes were generated by inserting individual Env genes into the HIV-1 molecular clone pNL4-3. Derivatives of these constructs expressing nanoluciferase (HIV-1 (nLuc) viruses) were generated by inserting the nLuc encoding sequences in place of Nef. HIV-1 viral stocks were generated by transfecting 293 T cells; supernatant was harvested 48 hr post transfection, filtered (0.2 µm), and titered on TZM-bl cells using a nanoluciferase assay.

## Cell lines

Cells (293T, ATCC CRL-3216) were stably transduced with a retroviral vector (LHCX) into which was inserted sequences encoding human CD4 and CCR5 genes separated by an FMDV 2A site. Single cell clones were selected and tested for CD4 and CCR5 expression by FACS analysis using Alexa-Fluor 647 anti-human CD4 (Biolegend) and PE anti-human CD195/CCR5 (BD Pharmingen). MT2/R5 cells were generated by transducing MT2 cells (NIH AIDS regent repository Catalogue number 237) with a retroviral vector encoding hCCR5 and selecting a single hCCR5+ cell clone. GHOSTX4 and GHOSTR5 cells, that express hCD4 and CXCR4 or CCR5, respectively, were obtained from the NIH AIDS reagent repository (Catalogue numbers 3685 and 3944), and subclones thereof were isolated by limiting dilution. Cells were monitored periodically for retrovirus contamination and were tested for mycoplasma and found to be negative. Identity of cell lines was verified by visual assessment of highly characteristic morphology and virus susceptibility.

## Neutralization and maraviroc assays

Serial dilutions of Maraviroc (NIH AIDS Reagent Program), purified IgG from RhIV infected mice, or bNAbs (VRC01 (from Xueling Wu), 10E8, PG16, PG9 (NIH AIDS Reagent Program), 3BNC117, 10–1074 (from Michel Nussenzweig)) were incubated with virus for 1 hr at 37°C prior to the addition of TZM-bl cells. For neutralization assays against RhIV and all Maraviroc assays, TZM-bl cells were seeded the day before in 96-well Flat bottom plates (Falcon). For neutralization assays against HIV-1, cells were added in suspension to the virus/antibody mixture after the incubation period. After 4 hr (RhIV) or 48 hr (HIV-1) of infection, cells were washed twice with PBS before adding 50 µl of 1X Passive Lysis Buffer (Promega). Cell lysates were mixed with an equal volume of Nano-Glo Luciferase Assay Buffer and Substrate (Promega), incubated for at least 3 min at room temperature, then read using a Modulus II Microplate Multimode Reader (Promega).

## CD4+/CCR5+ Transgenic Mice

Sequences encoding human CD4 and CCR5 genes separated by an FMDV 2A site were inserted into a construct containing the regulatory elements for CD4-specific transgene expression (*Killeen et al., 1993*). The linearized transgene construct was injected into C57BL/6J embryos (Rockefeller University Transgenic Services Laboratory) to generate transgene lines C57BL/6J-Tg(Cd4-CD4,CCR5)A1Bsz (CD4/CCR5$_{HI}$), C57BL/6J-Tg(Cd4-CD4,CCR5)C18Bsz (CD4/CCR5$_{INT}$), and C57BL/6J-Tg(Cd4-CD4, CCR5)B4Bsz (CD4/CCR5$_{LO}$). Individual transgenic lines were maintained in a hemizygous state (Tg/0) in a C57BL/6J background and genotyped for the presence of the transgene by PCR using the following primers: RL413 GAACCTGGTGGTGATGAGAGCCACTCA and RL425 TGCTTGCTTTAACA-GAGAGAAGTTCGT. Selected transgenic lines (termed #A1, #C18, and #B4) that were chosen based on high, intermediate and low levels of CD4 expression respectively, were also crossed with C57BL/6J *Ifnar*1 knockout mouse line (MMRRC #32045) (*Müller et al., 1994*) to generate corresponding #A1$_{Ifnar1-/-}$, #C18$_{Ifnar1-/-}$ and #B4$_{Ifnar1-/-}$ mouse lines.

## Infection and monitoring of mice

Mice derived from C57BL/6 of both sexes were used, and housed under standard conditions prior to infection. Mice were moved to an ABSL-2 facility and were randomly ascribed to experimental groups prior to infection. Initial infections were done at 8 to 12 weeks. Mice were infected with RhIV stocks (10 to $10^5$ PFU in 500 µl DMEM) by intraperitoneal (i.p.) injection. Thereafter, blood was collected in EDTA coated tubes (Sarstedt) from the facial vein at the indicated timepoints, typically 1, 4,

7, 14 and 21 days after infection, and weekly thereafter for longer term experiments. Plasma was separated from cells and used for extraction of RNA or in ELISA while cells were processed for FACS analysis of cell populations. For analysis of tissues, mice were euthanized using carbon dioxide. Spleen, thymus, and lymph node tissue was removed and processed for RNA extraction or FACS analysis. In experiments involving serum transfer, previously infected donor animals were bled several times, 3 to 4 days apart, serum isolated from each bleed and pooled with serum from the same individual mouse. Thereafter, 200 µl of heat inactivated serum was injected subcutaneously (s.c.) into naïve animals one day prior to an i.p. RhIV challenge. For monoclonal antibody protection experiments, antibodies (50 µg to 1 mg) were diluted in PBS to a final volume of 200 µl and administered s.c. one day prior to an i.p. RhIV challenge. All animal studies were conducted in accordance with The Rockefeller University Institutional Animal Care and Use Committee (IACUC).

## RNA extraction and RT-qPCR

Viral RNA was extracted from 50 µl aliquots of mouse plasma using Trizol LS Reagent (Ambion). Phase separation steps were performed in MaXtract High Density tubes (Qiagen) and GlycoBlue (Invitrogen) was used as a coprecipitant. After drying, RNA pellets were resuspended in 50 µl RNase-free Molecular Biology Grade Water (Corning). For RT-qPCR, 8 µl of purified RNA solution was used, and RT-qPCR were carried out in one step using the Power SYBR Green RNA-to-CT 1-Step kit (Applied Biosystems) and primers RL509 TGATACAGTACAATTATTTTGGGAC and RL510 GAGACTTTCTGTTACGGGATCTGG, that target the VSV-L gene (*Hole et al., 2006*). Duplicate aliquots of RNA were tested using an Applied Biosystems Step-One Plus Real Time PCR machine. A standard curve, generated using a plasmid DNA template, was used to calculate RNA copies/ml. The limit of detection for this assay was a single copy of cDNA per PCR reaction, equivalent to 125 RNA copies/ml of mouse plasma.

## Flow cytometry

Mouse blood, spleen, thymus and lymph node were processed for FACS analysis by making a single cell suspension, removing red blood cells by resuspension in red blood cell lysis buffer (150 mM $NH_4Cl$, 10 mM $KHCO_3$, and 0.05 mM EDTA), then resuspending the resulting pellet in FACS buffer (PBS, 0.2% bovine serum albumin). Cells were incubated with anti-CD16/CD32 (Fc Block) prior to staining with the following antibodies: FITC anti-CD3, PerCP-Cy5.5 anti-mCD4, PE anti-CD19 (BD Pharmingen), APC anti-CD8a, and APC/Cy7 anti-hCD4 (BioLegend). Samples were run on either a LSRII (Becton Dickinson) or Attune NxT (Life Technologies) flow cytometer and data were analyzed using FlowJo (Tree Star).

## IgG purification

For measurement of neutralizing activity in mouse serum, purified IgG was used. Serum was separated from whole blood by centrifugation and heat inactivated for 1 hr at 56°C. IgG was purified from serum using the Protein G HP Spin Trap/Antibody Spin Trap kit (GE Healthcare), according to manufacturers instructions then dialyzed overnight in PBS at 4°C (Slide-A-Lyzer, 20,000 MWCO, Thermo Scientific). Purified IgG solutions were then filtered through a 0.2 µm filter and concentrated (Spin-X UF Concentrator, Corning).

## Western blotting

GHOST R5 cells infected with RhIV were lysed with RIPA buffer (150 mM NaCl, 50 mM Tris, pH 7.4, 0.1% SDS, 1 mM EDTA, 1% Igepal, 1% sodium deoxycholate). Virions were pelleted through 20% sucrose in PBS. Cell and virion proteins were separated on SDS-PAGE gels and blotted onto nitrocellulose membranes. Blots were probed with anti-gp120 (American Research Products) and anti VSV-M (Kerafast) with IR800 donkey anti goat and IR680 donkey anti mouse (LiCor) secondary antibodies.

## ELISA and other binding antibody assays

The *env* genes of HIV-1 strains were synthesized by GeneART (Thermofisher), in a modified form to generate C-terminally His-tagged soluble SOSIP.664 Env trimers. The previously characterized

SOSIP.664 trimers were derived from the clade A BG505 strain (PMID: 24068931), the clade B B41 strain (PMID: 25589637) and two clade C strains DU422 and ZM197M (PMID: 26372963).

The env cDNAs were inserted into pAAV-MCS and the resulting Env expression plasmids were transiently transfected into Expi293 cells using the serum free Expi293 Expression System (Life Technologies). Cell culture supernatants were collected at 5 days post-transfection, sterile filtered, and used as a source of Env proteins. Corning Costar 96-well EIA/RIA Plates were coated with anti-His-Tag Antibody (pAb, Rabbit, GenScript, US) at a concentration of 1 μg/ml in coating buffer (0.05 M Carbonate-Bicarbonate, pH 9.6) overnight. Unbound antibody was removed with wash buffer (50 mM Tris, 0.14 M NaCl, 0.05% Tween 20, pH 8.0) and the plates blocked (50 mM Tris, 0.14 M NaCl, 1% BSA, pH 8.0). SOSIP Env proteins were captured via their C-terminal His-Tags from cell culture supernatants at 37°C for 1 hr. After washing steps, serial 1:2 dilutions of heat-inactivated mouse serum or plasma samples (beginning a 1:300 dilution) were added to the plate and incubated at 37°C for 2 hr. The plates were washed and blocking buffer containing an anti-mouse HRP-conjugated antibody (GAM Ab97040 HRP, 1:20,000, Abcam) was added and incubated at 37°C for 30 min. Unbound antibodies were removed by additional washing steps and bound HRP detected by TMB One Solution System (Promega). After 20 min of incubation the colorimetric reaction was stopped by adding 0.3M phosphoric acid and spectrophotometric readings recorded at 450 nm.

Human bNAbs were titrated to give ELISA signals on BG505 SOSIP.664 coated plates that were in the linear range with respect to bNAb concentration. For competition ELISAs, BG505 SOSIP.664 coated plates were preincubated for 2 hr with dilutions of IgG that had been purified from mouse sera. Then, the plates were washed and incubated with blocking buffer containing the predetermined concentrations of human bNAbs for 30 min. After washing, plates were incubated with blocking buffer containing an anti-human HRP-conjugated antibody (Ab97175 HRP, 1:20,000, Abcam) and bound antibodies detected as above.

For detection of antibodies using INNO-LIA HIV I/II Score strips, the manufacturers (Fujirebio) procedures were followed, except that Ab97175 goat anti-Hu IgG (HRP) 1:20 000 and Ab97040 goat anti-Ms IgG (HRP) 1:20 000 were used, as appropriate. Bound antibodies were detected using SuperSignal West Pico PLUS chemiluminescent substrate (Thermofisher).

## Replicates and statistics

All data is plotted raw, that is individual values for each individual determination and each individual mouse is plotted. The exceptions to this are the qRT-PCR data, in which the mean of technical duplicates is plotted. Animals were allocated randomly to experimental groups. Statistical comparisons between groups in *Figure 7D,E,F* were done using Graphpad Prism software, and p-values were calculated using a Mann Whitney test.

## Acknowledgements

We thank Dan Littman, Michel Nussenzweig, and Xueling Wu for reagents, and members of the Bieniasz laboratory for advice. This work was supported by grants from the NIH R01AI50111 and R37AI64003 (to PDB) and R01AI078788 (to TH).

## Additional information

### Funding

| Funder | Grant reference number | Author |
| --- | --- | --- |
| National Institute of Allergy and Infectious Diseases | R37AI064003 | Paul D Bieniasz |
| National Institute of Allergy and Infectious Diseases | R01AI078788 | Theodora Hatziioannou |
| National Institute of Allergy and Infectious Diseases | R01AI50111 | Paul D Bieniasz |
| Howard Hughes Medical Institute | | Paul D Bieniasz |

The funders had no role in study design, data collection and interpretation, or the decision to submit the work for publication.

## Author contributions
Rachel A Liberatore, Conceptualization, Investigation, Methodology, Writing—review and editing; Emily J Mastrocola, Conceptualization, Formal analysis, Investigation, Writing—review and editing; Elena Cassella, Jessie R Willen, Dennis Voronin, Trinity M Zang, Investigation; Fabian Schmidt, Conceptualization, Investigation, Methodology; Theodora Hatziioannou, Supervision, Project administration, Writing—review and editing; Paul D Bieniasz, Conceptualization, Formal analysis, Supervision, Funding acquisition, Writing—original draft, Project administration

## Author ORCIDs
Paul D Bieniasz (iD) https://orcid.org/0000-0002-2368-3719

## Ethics
Animal experimentation: This study was performed in strict accordance with the recommendations in the Guide for the Care and Use of Laboratory Animals of the National Institutes of Health. All of the animals were handled according to approved institutional animal care and use committee (IACUC) protocol (18047-H) of the Rockefeller University. All surgery was performed under anesthesia, and every effort was made to minimize suffering.

## Decision letter and Author response
Decision letter https://doi.org/10.7554/eLife.49875.SA1
Author response https://doi.org/10.7554/eLife.49875.SA2

## Additional files
### Supplementary files
• Transparent reporting form

### Data availability
All data generated or analysed during this study are included in the manuscript and supporting files.

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
