## [Decision Letter]

**Acceptance summary:**

We think that the small animal model you developed will be useful for the study of HIV-1 entry and anti-Env antibodies in vivo. It is interesting that the chimeric vesicular stomatitis virus constructs encoding various clade A, B and C HIV-1 Env proteins are replication competent and cause CD4^+^ T cell depletion in transgenic mice expressing human CD4 and the coreceptor CCR5. Your work shows that this new animal model recapitulates some aspects of acute HIV-1 infection in humans. It will be of interest for everybody interested in HIV-1 neutralization and entry and hopefully support development of effective and safe AIDS vaccines.

**Decision letter after peer review:**

Thank you for submitting your article "Rhabdo-immunodeficiency virus, a murine model of acute HIV-1 infection" for consideration by *eLife*. Your article has been reviewed by three peer reviewers, including Frank Kirchhoff as the Reviewing Editor and Reviewer #1, and the evaluation has been overseen by Tadatsugu Taniguchi as the Senior Editor. The following individual involved in review of your submission has agreed to reveal their identity: Welkin E Johnson (Reviewer #2).

The reviewers have discussed the reviews with one another and the Reviewing Editor has drafted this decision to help you prepare a revised submission.

Summary:

Liberatore et al. describe a novel and potentially useful small animal model for the study of HIV-1 entry and anti-Env antibodies in vivo. Specifically, they engineered replication-competent vesicular stomatitis virus (VSV)/HIV Env chimeras (RhIVs) by replacing the G gene of VSV with sequences encoding the ecto and transmembrane domains of various clade A, B and C HIV-1 Env proteins fused to the cytoplasmic tail of VSV G. Infection of hCD4/hCCR5 transgenic mice with these RhIVs resulted in acute viral loads in plasma and transient CD4^+^ T cell depletion reminiscent of HIV-1 infection. RhIVs display appropriate coreceptor tropism, are sensitive to the CCR5 inhibitor Maraviroc and induce Env-specific antibody responses in vivo. The chimeric viruses replicated more efficiently and elicited higher Env-specific antibody responses in IFNAR-/- Tg mice compared to IFNAR+/+ Tg mice. The CD4 dependence of the RhIVs mirrored their corresponding HIV-1 Envs as reflected by differences in their ability to replicate in transgenic mice with high versus low levels of CD4. Passive transfer of two HIV-1 bNAbs (PG16 and 3BNC117) resulted in dose-dependent protection against RhIV challenge, consistent with similar experiments in NHPs. RhIV infection conferred robust protection against reinfection with RhIVs expressing homologous or heterologous Envs; however, this protection did not require B cells as indicated by experiments in *µMT-/-* mice and with a VSV recombinant expressing ecotropic MLV Env (VSV MLV-E). Repeated RhIV infection elicited antibodies in serum that were capable of binding to SOSIP trimers by ELISA, but with the exception of weak stain-specific neutralization of HIV-1_SF162_, did not neutralize Env-matched HIV-1 isolates. Finally, convalescent sera from mice repeatedly infected with RhIV was tested for the ability to confer protection by passive transfer. Although mice that received sera from animals with weak neutralizing activity against HIV-1_SF162_ had lower peak viral loads after RhIV_SF162_ challenge, none were completely protected. In contrast, some of the mice that received sera from mice exposed to different combinations of RhIVs that lacked detectable neutralizing activity were protected against RhIV_SF162_ challenge.

Overall, this is a highly innovative study that offers a novel model for investigating antibody responses to HIV-1 Env in mice. As discussed by the authors, the model has a number of limitations, such as lack of persistence and rapid clearance. However, these caveats are discussed and the RhIV platform is of significant interest since it potentially allows for in vivo assessment of both antibody-mediated neutralization and antibody-mediated effector mechanisms, pre-testing of vaccine immunogenicity, and in vivo testing of small molecule entry inhibitors. The conclusions of the study are supported by an extensive, high-quality dataset and the manuscript is well written.

Essential revisions:

1) There is one caveat that should be straightforward to address – the study lacks a profile of the binding antibodies that are produced by infection, and how that basic profile compares to published HIV serum profiles. As the authors point out, their use of chimeric HIV Envs with the VSV-G cytoplasmic tail may have altered the arrangement of the extracellular domains. In combination with the binding of SOSIP proteins (already done and displayed in Figure 6—figure supplements 3-5), serum western blots and/or peptide ELISA would reveal whether there is a similar distribution of binding specificities to HIV infection, or whether the mice are mounting responses against just one or a few targets or targets that completely differ from HIV infection. The result would not change the importance of the study, but would potential alert adopters of the RhIV platform that additional optimization might be necessary.

2) For the most part, the results are adequately presented and discussed. Some statements, however, should be tempered. For example, the statement in the summary that the RhIV model "faithfully represents HIV-1 entry, tropism and antibody sensitivity". As the authors well know, the viral cell tropism is determined by many factors in addition to entry, and the observation that IFNAR1 deficient mice are much more susceptible to RhIVs suggest that ISGs also have a significant impact on the tropism of the chimeric viruses. The authors show that the transgenic mice express human CD4 and CCR5 at levels "precisely mimicking those on human CD4^+^ T-cells", but it would be of interest to know a bit more about the phenotype and state of activation of these cells.

3) As stated by the authors, the genetic manipulability of the mice and the chimeric virus will allow further studies on the elicitation of HIV-1 Env-specific antibodies. However, the rapid clearance of the infection limits the value of this model; e.g. examining the evolution of antibody specificities over time, may not be feasible or will require additional boosting (perhaps with divergent RhIV backbones). Are there possible strategies to delay viral replication and hence potentially clearance in this model? Given the almost complete loss of CD4^+^ T cells, do the authors think the infection is cleared by the immune response to infection or due to target cell exhaustion?

---

## [Author Response]

Essential revisions:1) There is one caveat that should be straightforward to address – the study lacks a profile of the binding antibodies that are produced by infection, and how that basic profile compares to published HIV serum profiles. As the authors point out, their use of chimeric HIV Envs with the VSV-G cytoplasmic tail may have altered the arrangement of the extracellular domains. In combination with the binding of SOSIP proteins (already done and displayed in Figure 6—figure supplements 3-5), serum western blots and/or peptide ELISA would reveal whether there is a similar distribution of binding specificities to HIV infection, or whether the mice are mounting responses against just one or a few targets or targets that completely differ from HIV infection. The result would not change the importance of the study, but would potential alert adopters of the RhIV platform that additional optimization might be necessary.

Clearly, the serological response to HIV-1 infection can vary considerably between human individuals (for example, only a small percentage generate bnAbs to one of several epitopes). Thus, it is quite difficult to compare the detailed (i.e. epitope specific) serological response in ‘mice’ compared to ‘humans’ without detailed analysis of cohorts in both species. Nevertheless, we did some analysis using commercially available nitrocellulose strips loaded with HIV-1 antigens, as well as competition ELISAs. These data show that RhIV infected mice (i) generate antibodies to both gp41 and gp120 (ii) the antibodies generated do not compete with sCD4 or bnAbs from a small panel the recognize the CD4bs, the V2 apex and the V3 loop. These data are presented in a new supplementary figure (Figure 6—figure supplement 6) and described in the first paragraph of the subsection “Serological responses to HIV-1 Env in RhIV-infected mice”. The details of the serological response in RhIV infection are obviously of interest, but we feel that a more detailed analysis of the serological profile would go beyond the scope of the study and would require considerable additional effort.

2) For the most part, the results are adequately presented and discussed. Some statements, however, should be tempered. For example, the statement in the summary that the RhIV model "faithfully represents HIV-1 entry, tropism and antibody sensitivity". As the authors well know, the viral cell tropism is determined by many factors in addition to entry, and the observation that IFNAR1 deficient mice are much more susceptible to RhIVs suggest that ISGs also have a significant impact on the tropism of the chimeric viruses. The authors show that the transgenic mice express human CD4 and CCR5 at levels "precisely mimicking those on human CD4^+^ T-cells", but it would be of interest to know a bit more about the phenotype and state of activation of these cells.

We have gone through the manuscript and tempered statements in instances where our language may have been over-enthusiastic about the accuracy and utility of this model system, including the specific instances mentioned by the reviewers.

Specifically,

“Faithfully represents” becomes “recapitulates key features of”

“precisely” was deleted

“nearly precisely” was deleted

“Faithfully represents” becomes “recapitulates key features of”

“faithfully recapitulated a normal pattern of CD4 expression” becomes “was restricted to T-cells that normally express mCD4”

3) As stated by the authors, the genetic manipulability of the mice and the chimeric virus will allow further studies on the elicitation of HIV-1 Env-specific antibodies. However, the rapid clearance of the infection limits the value of this model; e.g. examining the evolution of antibody specificities over time, may not be feasible or will require additional boosting (perhaps with divergent RhIV backbones). Are there possible strategies to delay viral replication and hence potentially clearance in this model? Given the almost complete loss of CD4^+^ T cells, do the authors think the infection is cleared by the immune response to infection or due to target cell exhaustion?

We have given much thought to these issues. The reviewer is correct in that an optimal model would be one in which the RhIV persists, enabling co-evolution of Env and antibody to be studied. We have tried a number of approaches, such as insertion of HIV-1 accessory genes into RhIV stains and serial passage of RhIV strains in mice to achieve persistent replication, as yet without success. Alternatively, the ability to re-infect with HIV-1 enveloped viruses with different backbones would enable further ‘development’ of an antibody response. We have added and modified some text in the Discussion to highlight limitations of the current model and the potential utility of other viruses with HIV-1 envelopes as ‘boosters’ that would potentially enable the evolution of the HIV-1 antibody response to be studied (Discussion, fourth paragraph).

We do not think (although this is opinion – not necessarily fact) that the infection is cleared due to target cell exhaustion. Some of our attempts to make more persistent viruses have resulted in viruses that have reduced peak viremia, and incomplete CD4 T-cell depletion, yet have clearance kinetics that are essentially the same as those reported herein. We favor the hypothesis that a cell mediated adaptive response (the most obvious candidate being that mediated by CD8^+^ T-cells) is responsible for clearance and subsequent protection. We are currently investigating the role of CD8^+^ cells in clearance.